# Statistical Analysis and Scenario Modeling of Non-Stationary Runoff Change in the Loess Plateau: A Novel Application of the Generalized Additive Model in Location, Scale and Shape

**Shuqi Zhang [1], Tong Zhi [1], Hongbo Zhang [1,2,3,*], Chiheng Dang [1], Congcong Yao [1], Dengrui Mu [1], Fengguang Lyu [1], Yu Zhang [1] and Shangdong Liu [1]**

[1] School of Water and Environment, Chang'an University, Xi'an 710054, China; 2018029003@chd.edu.cn (S.Z.); 2018129009@chd.edu.cn (T.Z.); 2021029001@chd.edu.cn (C.D.); 2020029001@chd.edu.cn (C.Y.); 2020029008@chd.edu.cn (D.M.); 2023029007@chd.edu.cn (F.L.); 2022029001@chd.edu.cn (Y.Z.); 2023029013@chd.edu.cn (S.L.)

[2] Key Laboratory of Eco-Hydrology and Water Security in Arid and Semi-Arid Regions of the Ministry of Water Resources, Chang'an University, Xi'an 710054, China

[3] Key Laboratory of Subsurface Hydrology and Ecological Effect in Arid Region of Ministry of Education, Chang'an University, Xi'an 710054, China

*   Correspondence: hbzhang@chd.edu.cn

**Abstract:** The hydrological series in the Loess Plateau region has exhibited shifts in trend, mean, and/or variance as the environmental conditions have changed, indicating a departure from the assumption of stationarity. As the variations accumulate, the compound effects caused by the driving variables on runoff variations grow complex and interactive, posing a substantial risk to water security and the promotion of high-quality development in regions or river basins. This study focuses on the Tuwei River Basin in the Loess Plateau, which experiences significant changes in vegetation coverage and minimal human disturbance, and examines the cross-driving relationship between the runoff change and its driving variables (including hydrometeorological and environmental variables). A quantitative statistical analysis method based on the GAMLSS is then developed to estimate the interacting effects of changes in the driving variables and their contribution to runoff changes. Finally, various anticipated scenarios are used to simulate the changes in driving variables and runoff disturbances. The findings indicate the following: (1) The developed GU, LO, and NO distribution-based GAMLSSs provide a notable advantage in effectively capturing the variations in groundwater storage variables, actual evapotranspiration, and underlying surface parameters, as well as accurately estimating the impacts of other relevant variables. (2) The precipitation and groundwater storage variables showed predominantly positive contributions to the runoff change, but actual evapotranspiration had an adverse effect. The changes in underlying surface parameters, particularly since 2000, increase actual evapotranspiration, while decreasing groundwater storage, resulting in a progressive decrease in runoff as their contribution grows. (3) The scenario simulation results reveal that alterations to the underlying surface have a substantial influence on the evolution of runoff in the Tuwei River Basin. Additionally, there are cross-effects between the impact of various driving variables on runoff, potentially compounding the complexity of inconsistent changes in runoff sequences.

**Keywords:** hydrological non-stationarity; attribute analysis; scenario modeling; GAMLSS; Tuwei River Basin

## 1. Introduction

Precipitation, evaporation, water storage, and the underlying surface condition are commonly regarded as the primary causes of watershed runoff. Recently, they have seen varying degrees of change, impacted by increasingly intense climate change and human

activities around the world, primarily generating non-stationary changes in runoff in arid and semi-arid regions [1,2]. Furthermore, the elements impact each other, amplifying their interactions and driving effects on runoff in these regions [3–5]. The Loess Plateau (LP) comprises the largest area, and it is receiving increased attention, particularly since the vegetation cover has altered dramatically [6–8]. As widely described in the literature, the major elements in the LP's water cycle process have changed dramatically in recent years as a result of climate change and human-induced vegetation regreening [9]. Runoff, as a result of changes in numerous elements, is heavily influenced by the compound impacts of these changes [7,10,11]. Physically, precipitation, evapotranspiration, and groundwater storage have a direct impact on runoff changes, whereas changes in the underlying surface are the primary environmental factors that indirectly cause runoff changes. However, when the streamflow is non-stationary, it is extremely difficult to disentangle the contributions of various elements to the inconsistent changes in runoff to figure out the attribution explanations of significant variations in runoff on the LP.

As a result, many scientists have explored the inconsistency and attribution of regional hydrological processes using various time series models or physical hydrological models. For example, Feng et al. (2016) used a multiple regression model to investigate the interactive roles of climate and human activities on runoff decline in 14 basins in the LP and discovered that reduced precipitation was the primary reason for the decrease in runoff between 1961 and 2009, with human intervention playing a dominant role in producing these shifts, after which the water yield decreased further [6]. Zhang et al. (2020) used the partial least squares regression (PLSR) approach to evaluate the contribution of the expanding implementation of ecological restoration (ER) strategies in the LP to achieve streamflow decline, and the results revealed that ER was the dominant cause of streamflow reduction, with the contribution increasing from 59% in 1980–1999 to 82% in 2000–2015 [10]. Tan et al. (2024) proposed a modified Budyko attribution method to quantify vegetation-induced runoff alterations in the LP, and the findings show that the vegetation change mainly caused runoff reduction over the LP, resulting in 78.94% of the reduced runoff, and the "Grain-for-Green" Program (GFGP)-led LULC shift, particularly for cropland reduction, plays a vital role in vegetation-induced runoff losses, which could increase future water stress in the LP [7]. Gao et al. (2020) used the SWAT model to simulate runoff change under several scenarios in the Jing River Basin of the LP and evaluated the climatic and anthropogenic impacts. The results showed that the impact of climatic elements progressively diminished over time, while the influence of direct variables (water withdrawal) expanded the fastest, and the influence of indirect causes steadily increased [12]. Sun et al. (2019) used the RCC-WBM model to quantitatively separate the impacts of climate change and human activities on runoff change in the Tao River from the Tibetan Plateau to the LP and found that human activities are the primary drivers of runoff reduction in the Basin, though both these absolute influences tend to increase [13]. Liu et al. (2012) used the Tsinghua Hydrological model based on the Representative Elementary Watershed approach (THREW) to investigate the characteristics of runoff generation in the LP, concluding that the subsurface flow contribution to total streamflow is greater than 53% from October to March, while the overland flow contribution exceeds 72% from April to September [14].

Although the aforementioned research has made significant progress in interpreting the interactive effects of the hydrological cycle on the LP, as well as analyzing the attribution of non-stationary characteristics in runoff sequences, these two models or methods have some limitations. For example, time series analysis can only provide average and general contribution estimates, such as from climate change, ecological restoration (ER) strategies, or human activities. In other words, while these methods may be beneficial for analyzing non-stationary time series, their intrinsic static regression aspect does not properly describe many complicated physical interaction processes [15]. Hydrological models, such as lumped (IHACRES), semi-distributed (HEC-HMS), and fully distributed (SWATgrid) ones, accurately simulate runoff in smaller catchments under most hydroclimatic conditions, but they frequently fail in larger catchments, regardless of the hydroclimatic conditions [16],

particularly in catchments with non-stationarity in rainfall–runoff relationships. This implies the need for additional research to promote the establishment of the cross-driving relationship between runoff non-stationary changes and their driving elements, as well as the quantitatively evaluation of variations in the driving elements and their impact on runoff alterations, both of which are critical in formulating water resource management policies and addressing water shortages on the LP.

The Generalized Additive Model in Location, Scale and Shape (GAMLSS) is a tool for modeling time series under non-stationary conditions [17]. It supports a variety of random variable frequency distribution types and is extremely useful in constructing linear or nonlinear functional relationships between distribution function position parameters, scale parameters, shape parameters, and explanatory variables [18]. The GAMLSS framework has been widely applied in non-stationary frequency analysis, modeling, and forecasting in hydrology [19–22]. This GAMLSS feature also allows for cross-driving interactions between runoff and the driving elements, or between the driving elements themselves. In light of this, the goal of this paper is to create an inconsistent hydrological statistical model (GAMLSS) with physical factors as covariates in order to identify the cross-driving relationship between watershed hydrometeorological (precipitation, actual evapotranspiration, groundwater storage variables, and runoff) and surface environmental variables (underlying surface parameters) with inconsistently changing characteristics. It offers a novel approach to inquiry that differs from the previous time series analysis and hydrological models in that it statistically analyzes the cross-driving relationship between elements and investigates the causes of runoff decline.

The study entails (1) investigating the cross-driving relationship between runoff changes and their driving delivers (including hydrometeorological and surface environmental elements), (2) developing a hydrological non-stationary model that incorporates several driving elements, (3) quantitatively examining the changes in runoff driving variables and their impact on runoff evolution, and (4) developing scenario plans to simulate future runoff change patterns under various driving impacts. Although this study focuses on the Loess Plateau, the findings are extremely relevant to water managers in other arid and semi-arid regions with substantial hydroclimatic fluctuations or changes.

## 2. Study Area and Data

The Tuwei River is a first-level tributary of the Yellow River. It is situated in Yulin City, in the northern part of Shaanxi Province, China [23]. The river spans 140.0 km, covers 3294.0 km$^2$ of drainage area [24], and has an average channel ratio of 3.87‰. The Tuwei River's water system is simple, with a branch-like distribution. The tributaries along the southwest bank are more developed [25]. The largest tributaries include the Qingshui, Zhalinchuan, Yangjiapan, and Kaiguangchuan Rivers. The Tuwei River Basin has a multi-year average runoff of around 380 million m$^3$. A significant regional variance in runoff is visible. The upstream level is substantially higher than the downstream level, and the runoff recharge is primarily groundwater. Seasonal variations include two flood periods: a spring flood and a summer flood. The river has a low silt composition, with an annual sediment transit volume of 33.5 million tons, with the highest amount occurring between June and September. The Tuwei River Basin has a continental monsoon climate, which means hot summers and cold winters. The multi-year average temperature is 8.5 °C, with an average precipitation of 392.2 mm. Rainfall is irregularly distributed throughout the year, with the bulk falling from June to September. The basin's multi-year average wind speed is 2–3.6 m/s, and the average number of sunlight hours is 2853.

The watershed hydrometeorological observations include precipitation, potential evapotranspiration, and runoff. Since there is no national meteorological station in the Tuwei River Basin, potential evapotranspiration was calculated using data from the Shenmu meteorological station, which is close to the basin. The meteorological data came from China's Meteorological Data Network (http://data.cma.cn/). The precipitation data are collected from seven rainfall stations in the basin and cover the period 1957–2010. The statistics

are from the Yellow River Basin Hydrological Data Yearbook. Also, the Thiessen polygon method was used to acquire surface precipitation data on the watershed. Gaojiachuan Station serves as the Tuwei River Basin's discharge hydrological station. The station's daily flow data from 1957 to 2010 was utilized. These data also came from the Yellow River Basin Hydrological Data Yearbook. Figure 1 depicts the distribution of meteorological, rainfall, and hydrological stations throughout the basin.

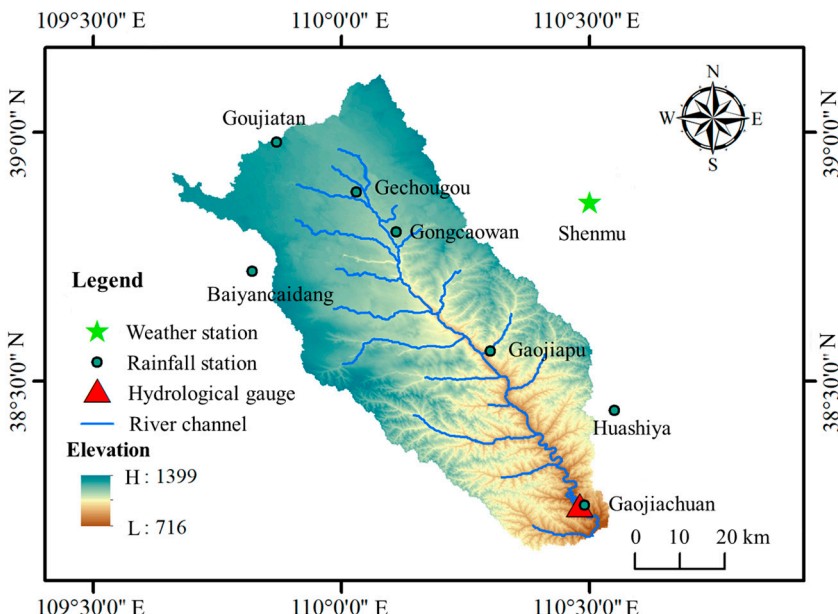

**Figure 1.** Distribution of meteorological, rainfall, and hydrological stations throughout the Tuwei River Basin.

## 3. Methodology

The basin's yearly scale water balance formula (P = E + ET + ΔG, when surface water storage changes a little) reveals that precipitation, actual evapotranspiration, and groundwater storage variables are the principal drivers of runoff changes. The underlying surface conditions have a significant impact on actual evapotranspiration and groundwater storage variables, making them crucial concerns that cannot be overlooked when investigating the evolution of the runoff process in changing environments. As a result, this study selected precipitation, actual evapotranspiration, groundwater storage variables, and the parameter *n* in the Budyko equation [26], which represents changes in the underlying surface, as the key driving forces impacting runoff variations. Therein, groundwater storage variables were obtained using the USGS RORA model [27,28]. Also, an inconsistent GAMLSS was developed by identifying and establishing the interaction of runoff changes and their physical driving elements (e.g., groundwater storage variables, actual evapotranspiration, and underlying surface characteristics) as covariates [29]. This model studies how non-stationary fluctuations in yearly precipitation, groundwater storage factors, and actual evapotranspiration affect annual runoff. Finally, using the planned scenario, runoff variations caused by multiple diverse driving forces were simulated, and alternative co-evolution rules were investigated. Therein, the first year of the entire series (1957) was selected as the base year of evolution analysis.

### 3.1. Interaction of Runoff and Physical Driving Elements

Figure 2 depicts the interactive relationship and influence of runoff and its underlying variables. Precipitation, as a component of watershed moisture conditions, has a direct impact on groundwater storage variables, evapotranspiration, underlying surface parameters, and runoff. Precipitation, underlying surface factors, and actual evapotranspiration all interact to drive groundwater storage variables synergistically. When precipitation

reaches the aquifer, the amount of groundwater recharge increases, thereby increasing groundwater storage. As the underlying vegetation condition improves, the amount of infiltration increases, resulting in increases in groundwater recharge and groundwater storage. When actual evapotranspiration increases, groundwater recharge decreases, and groundwater storage shrinks. A mix of precipitation and surface properties influences actual evapotranspiration. When precipitation rises, so does actual evapotranspiration; when the underlying vegetation condition improves, so does transpiration from vegetation. Precipitation is what drives the underlying surface characteristics. When precipitation increases, plants absorb more water to meet their growth needs, resulting in improved vegetation conditions and, as a result, an increase in the underlying surface parameter $n$.

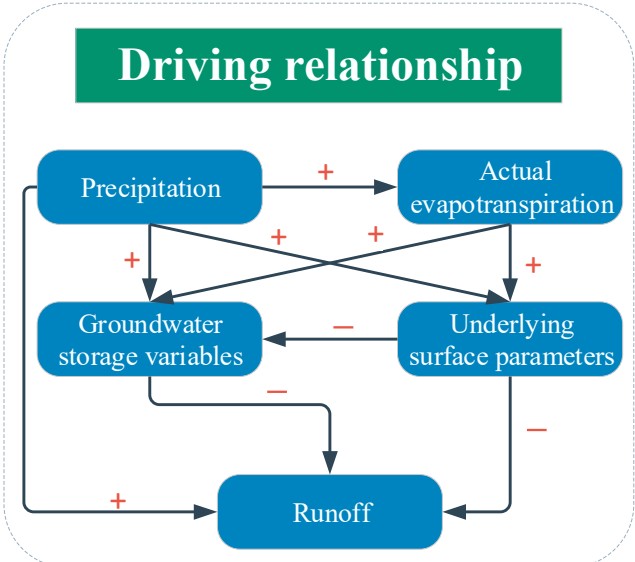

**Figure 2.** Interaction relationship between runoff and its physical driving elements. "+" represents positive driving effect, and "−" represents opposite driving effect.

*3.2. GAMLSS*

A Generalized Additive Model in Location, Scale, and Shape (GAMLSS) is a semi-parametric regression model that analyzes the frequencies of stationary and non-stationary runoff and other features [17–22,29,30].

For any time $t(t = 1, 2, \cdots n)$, the GAMLSS assumes that the distribution function of the mutually independent random variable observation sequence $y_t(t = 1, 2, \cdots n)$ is $F_y(Y_t | \theta^t)$ and the probability density function is $f(y_t | \theta^t)$, where $\theta^t = (\theta_{t1}, \theta_{t2}, \cdots \theta_{tm})$ is the set of m parameter vectors (including position, shape, and scale parameters) related to explanatory variables (covariates) and random effects. $k = 1, 2, \cdots, m, g_k(\cdot)$ is defined the monotonic connection function between the parameter vector $\theta_k$, the explanatory variables, and the random effect terms. If the random effect term is ignored, the expression is as follows:

$$g_k(\theta_k) = X_k \beta_k \tag{1}$$

where $X_k$ is an explanatory variable matrix with dimension $n \times p$, and $\beta_k$ is a parameter vector with length $p$.

Usually, the GAMLSS has no more than four distribution parameters. Most distributions have only two parameters: a position parameter and a scale parameter. The position parameter $\beta_1$ is regard as the first parameter vector $\theta_1$, representing the mean value $\mu$ of the random variable, whereas the scale parameter $\beta_2$ refers to the second parameter vector $\theta_2$, representing the mean square error σ of the random variable. Shape parameters $\beta_3$, such

as skewness $\nu$ and kurtosis $\tau$, are also included in more complicated distributions. If the distribution function has three parameters, the equation can be stated as follows:

$$
\begin{aligned}
g_1(\mu) &= X_1\beta_1 \\
g_2(\sigma) &= X_2\beta_2 \\
g_3(\nu) &= X_3\beta_3
\end{aligned}
\tag{2}
$$

The explanatory variable can be stated in matrix form as follows:

$$
X_k = \begin{bmatrix}
1 & x_{11} & \cdots & x_{1pk} \\
1 & x_{21} & \cdots & x_{2pk} \\
\vdots & \vdots & \ddots & \vdots \\
1 & x_{n1} & \cdots & x_{npk}
\end{bmatrix}
\tag{3}
$$

The maximum likelihood method is used to determine the model parameters. The likelihood function is as follows:

$$
L(\beta_1, \beta_2, \beta_3) = \sum_{t=1}^{n} ln(F(y_t|\beta_1, \beta_2, \beta_3))
\tag{4}
$$

where $y_t$ is the measured value of the sequence, $n$ is its length or the sample size, and $F$ is the cumulative probability distribution function that the series follows. However, it should be noted that the L-moments method is recommended for use when less than 50 data values are collected [30].

This study used the AIC (Information Criterion) and SBC (Schwartz Bayesian Information Criterion) to assess the appropriateness of model fitting. The lower the *AIC* and *SBC* values are, the better the model fits. As a result, the optimal model in this study is the one with the lowest *AIC* and *SBC* values. The *AIC* and *SBC* values are calculated using the following formulas:

$$
AIC = -2lnL\,(\beta_1, \beta_2, \beta_3) + 2df
\tag{5}
$$

$$
SBC = -2lnL\,(\beta_1, \beta_2, \beta_3) + dfln(n)
\tag{6}
$$

where $lnL(\beta_1, \beta_2, \beta_3)$ is the log-likelihood function associated with the regression parameter estimate, and $df$ is the log-likelihood function's degree of freedom.

Also, the residual distribution of the fitted model is an important criterion for assessing the model's fit. As a result, a worm plot is used in this study to determine if the model's residual sequence follows a normal distribution. If the residual sequence falls within the upper and lower bounds of the theoretical curve, it is assumed that the residual sequence follows the normal distribution and that the model fits correctly. In addition, this study employs the mean, variance, skewness coefficient, kurtosis coefficient, and Filliben correlation coefficient as statistical markers of the residual sequence. The mean is closer to 0, the variance is closer to 1, the skewness coefficient is closer to 0, the kurtosis coefficient is closer to 3, and the Filliben coefficient is closer to 1, indicating that the model fits better [31].

Initially, the consistency analysis of each variable sequence in this study was carried out. Six two-parameter distributions were chosen, Gumbel/GU, normal/NO, logistic/LO, gamma/GA, log normal/LOGNO, and Weibull/WEI, which are commonly employed in the frequency analysis of extreme events around the world [32,33], particularly in North China [34,35]. Their fit was tested using the GAMLSS by fixing the stable model's parameters, with the goal of determining the best appropriate consistency probability distributions for each sequence. The three most suitable distributions of each sequence were then submitted to non-consistency analysis with time as a covariate, and the parameters in the GAMLSS were adjusted over time to examine the trend of each sequence parameter changing over time. The sequence is judged to have undergone non-consistent changes if the *AIC* and *SBC* values of the GAMLSS with time as a covariate are less than those of the consistency model. Finally, the non-stationary GAMLSS of runoff with physical factors as

covariates is used to investigate non-consistent variations in runoff due to the influence of physical driving variables.

### 3.3. Quantitative Statistical Simulation of Observed Variable Changes

Taking into account the driving relationship between the various variables that have been constructed, this study uses the GAMLSS with physical influencing factors as covariates to conduct non-consistency analysis on the observed variable [29], and then the relationship expression of the observed variable's distribution parameters with the change in physical factors is obtained. The formula is as follows:

$$y^i = F\left(y \,\middle|\, \left(x_1^i, x_2^i, \ldots X_n^i \middle| \beta_1, \beta_2, \beta_3\right)\right) \tag{7}$$

where $y^i$ is the simulated value of the variable in the *i*-th year; $x_1^i, x_2^i, \ldots X_n^i$ is the value of each covariate in the *i*-th year; $F$ is the distribution function that the variable follows; and $\beta_1, \beta_2, \beta_3$ is the model's distribution parameter (i.e., position, scale, and shape parameters).

Each factor's covariate is $x_1, x_1, \ldots x_n$, and $\Delta y_1^i, \Delta y_2^i, \ldots \Delta y_n^i$ represents the covariate $x_1, x_1, \ldots x_n$'s contribution to factor $\Delta y$ in the *i*th year. The computation formula is given as follows:

$$\begin{cases} \Delta y_1^i = y^i\left(x_1^i, x_2^0, \ldots x_n^0\right) - y^0\left(x_1^0, x_2^0, \ldots x_n^0\right) \\ \Delta y_1^i = y^i\left(x_1^i, x_2^0, \ldots x_n^0\right) - y^0\left(x_1^0, x_2^0, \ldots x_n^0\right) \\ \vdots \\ \Delta y_n^i = y^i\left(x_1^i, x_2^i, \ldots x_n^i\right) - y^i\left(x_1^i, x_2^i, \ldots x_{n-1}^i, x_n^0\right) \end{cases} \tag{8}$$

where $x_1, x_1, \ldots x_n$ are the values of covariates in the base year (1957).

Finally, by summing the contributions of each covariate to the observed variable, it can be concluded that the simulated value of observed variable changes in year *i* versus 1957. The computation formula is given as follows:

$$\Delta y^i = \Delta y_1^i + \Delta y_2^i + \cdots \Delta y_n^i \tag{9}$$

where $\Delta y^i$ represents the simulated value of observed variable changes in year *i* as compared to that of the base year.

### 3.4. Scenario Setting

The baseline condition is a driving process that takes into account multi-year average precipitation (*P*). Furthermore, three simulation scenarios are defined, in which precipitation and underlying surface parameters (*n*) are changed from the baseline conditions, allowing for groundwater storage variables, actual evapotranspiration (*E*), underlying surface parameters, and runoff to be simulated. Furthermore, the evolution process and the change degree of each variable are obtained. The baseline conditions and three non-consistency scenarios are established as follows:

- S1: Baseline conditions are based on the multi-year average precipitation;
- S2: Precipitation increases by 10%;
- S3: Underlying surface parameters also increase by 10%;
- S4: Both precipitation and surface parameters increase 10% simultaneously.

## 4. Results and Discussions

### 4.1. Statistical Attribution of Changes in Groundwater Storage Variable

As we all know, the yearly groundwater storage variables are primarily determined by combining annual precipitation, actual evapotranspiration, and underlying surface characteristics [36]. As a result, a GAMLSS using yearly precipitation, annual actual evapotranspiration, and annual underlying surface characteristics as covariates can be developed. Specifically, this may be classified into the following three situations: (1) the mean

$\mu$ changes with the physical factors, and the mean square error $\sigma$ is a constant; (2) the mean $\mu$ is a constant, and the mean square error $\sigma$ changes with the physical factors; and (3) both the mean $\mu$ and mean square error $\sigma$ change with the physical factors. The minimum *AIC* can be used to determine the optimal distribution function of yearly groundwater storage variables under the effect of physical factors, as well as the relationship equation of distribution parameters that change with the physical factors.

4.1.1. Mean $\mu$ Changes with Physical Factors, and Mean Square Error $\sigma$ Is a Constant

A non-stationary model of yearly groundwater storage variables with physical factors as covariates was created, in which the mean ($\mu$) changes with the physical factors and the mean square error ($\sigma$) is constant. Table 1 provides the *AIC* values for the model. The table shows that when the mean $\mu$ varies with the physical factors and the mean square error $\sigma$ remains constant, the GU distribution, with annual precipitation, actual evapotranspiration, and underlying surface parameters as covariates, has the smallest *AIC* value.

**Table 1.** *AIC* values for the non-stationary GAMLSS of annual groundwater storage variable utilizing physical variables as covariates, with only the mean value $\mu$ changing.

| Covariate Combination | GU | LO | NO |
|:---:|:---:|:---:|:---:|
| *P* | 64.21 | 69.98 | 76.48 |
| *E* | 64.71 | 70.10 | 75.71 |
| *n* | 64.02 | 67.22 | 71.45 |
| *P* + *E* | −0.27 | −0.24 | −1.59 |
| *P* + *n* | 30.19 | 26.10 | 24.33 |
| *E* + *n* | 54.38 | 47.07 | 47.27 |
| *P* + *E* + *n* | −9.33 | −6.02 | −6.28 |

Table 2 presents the distribution parameters. The relationship between the mean $\mu$ and physical factors demonstrates that annual groundwater storage increases with precipitation, declines with evapotranspiration, and increases with the underlying surface characteristics.

**Table 2.** Distribution parameter estimates and goodness-of-fit test for GU-GAMLSS with only the mean value $\mu$ changing.

| Distribution | Covariate Combination | Position Parameter | Scale Parameter | *AIC* | *SBC* |
|:---:|:---:|:---:|:---:|:---:|:---:|
| GU | *P* + *E* + *n* | $0.02P - 0.03E + 2.15n - 1.76$ | $\log\sigma = -1.75$ | −9.33 | 0.62 |

4.1.2. Mean $\mu$ Is a Constant, and Mean Square Error $\sigma$ Changes with Physical Factors

A non-stationary model of yearly groundwater storage variables was created, in which the mean $\mu$ remained constant, whereas the mean square error $\sigma$ varied based on the physical factors. Table 3 provides the *AIC* values for the model, showing that almost all of the distributions have large *AIC* values. The only one that does not have very small *AIC* values is the NO distribution, which uses annual precipitation and underlying surface characteristics as covariates and only modifies the mean square error ($\sigma$) with the physical factors.

**Table 3.** *AIC* values for the non-stationary GAMLSS of annual groundwater storage variable utilizing physical variables as covariates; only the mean square error $\sigma$ varies with physical variables.

| Covariate Combination | GU | LO | NO |
|:---:|:---:|:---:|:---:|
| *P* | 64.77 | 70.13 | 76.69 |
| *E* | 63.92 | 70.13 | 76.69 |

**Table 3.** *Cont.*

| Covariate Combination | GU | LO | NO |
|:---:|:---:|:---:|:---:|
| $n$ | 61.66 | 58.56 | 61.40 |
| $P + E$ | 56.86 | 48.33 | 47.06 |
| $P + n$ | 55.33 | 47.34 | 45.80 |
| $E + n$ | 54.41 | 48.06 | 46.27 |
| $P + E + n$ | 56.02 | 49.34 | 47.79 |

The NO distribution parameters are presented in Table 4.

**Table 4.** Distribution parameter estimates and goodness-of-fit test for NO-GAMLSS with only the mean square error $\sigma$ varying.

| Distribution | Covariate Combination | Position Parameter | Scale Parameter | AIC | SBC |
|:---:|:---:|:---:|:---:|:---:|:---:|
| NO | $P + n$ | 0.29 | $\log\sigma = -0.01P + 2.43n - 1.30$ | 45.80 | 53.76 |

### 4.1.3. Both Mean $\mu$ and Mean Square Error $\sigma$ Change with Physical Variables

A non-stationary model was created, in which both the mean $\mu$ and mean square error $\sigma$ change with the physical variables. The *AIC* values for the model are presented in Table 5. The table demonstrates that when physical factors modify the mean $\mu$ and mean deviation $\sigma$, the GU distribution, with yearly precipitation, actual evapotranspiration, and underlying surface characteristics as covariates, has the lowest *AIC* value.

**Table 5.** AIC values for the non-stationary GAMLSS of annual groundwater storage variable utilizing physical variables as covariates; both mean $\mu$ and mean square error $\sigma$ change with physical variables.

| Covariate Combination | GU | LO | NO |
|:---:|:---:|:---:|:---:|
| $P$ | 64.77 | 70.13 | 76.69 |
| $E$ | 63.92 | 70.13 | 76.69 |
| $n$ | 61.66 | 58.56 | 61.40 |
| $P + E$ | 56.86 | 48.33 | 47.06 |
| $P + n$ | 55.33 | 47.34 | 45.80 |
| $E + n$ | 54.41 | 48.06 | 46.27 |
| $P + E + n$ | 56.02 | 49.34 | 47.79 |

The distribution parameters are reported in Table 6. The relationship between the mean $\mu$ and physical factors shows that annual groundwater storage increases with more precipitation, decreases with more evapotranspiration, and increases with larger underlying surface parameters.

**Table 6.** Distribution parameter estimates and goodness-of-fit test for GU-GAMLSS, with both mean $\mu$ and mean square error $\sigma$ changing with physical variables.

| Distribution | Covariate Combination | Position Parameter | Scale Parameter | AIC | SBC |
|:---:|:---:|:---:|:---:|:---:|:---:|
| GU | $P + E + n$ | $0.02P - 0.03E + 3.32n - 2.35$ | $\log\sigma = 0.02P - 0.03E + 6.01n - 4.85$ | $-11.45$ | 4.46 |

To better examine the research results, the annual groundwater storage variables are treated with physical factors as covariates, and the mean $\mu$ fluctuates with the physical factors, as does the mean square error $\sigma$. Figure 3 depicts the optimal non-consistency model's quantile plots at 5%, 50%, and 95%.

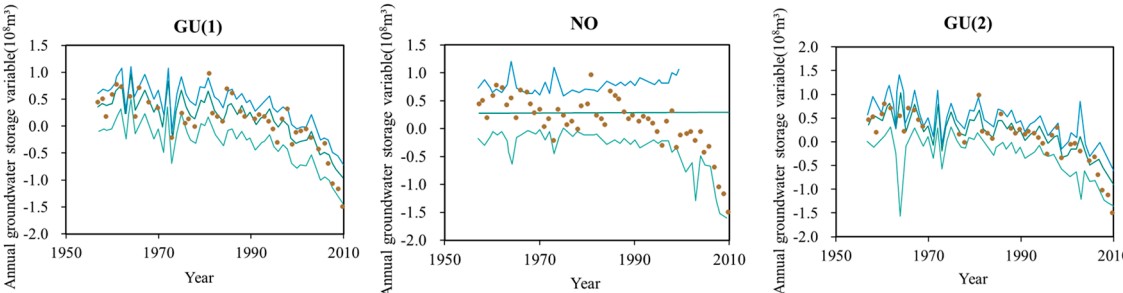

**Figure 3.** Five % (blue), fifty % (dark green), and ninety-five % (light green) quantile plots of the best non-stationary model for three distributions of annual groundwater storage variables, with *P*, *E*, and *n* as covariates.

Figure 4 and Table 7 show a worm plot, statistical indicators, and Filliben correlation coefficient for the residual sequence, respectively. The above chart shows that when the mean $\mu$ and mean deviation $\sigma$ change, owing to physical causes, the GU distribution, with yearly precipitation, evapotranspiration, and surface characteristics as covariates, has a better fitting impact. Furthermore, the *AIC* value is lower than those of the consistent model for yearly groundwater storage and the inconsistent model using time as a covariate. The residual sequences all fall inside the normal distribution's confidence interval, and the Filliben correlation coefficient exceeds 0.95, indicating that the inconsistent changes in the yearly groundwater storage variable are better conveyed.

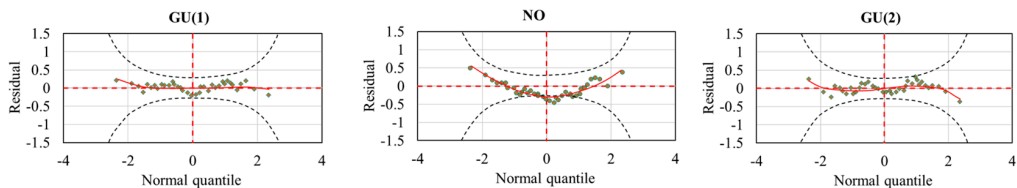

**Figure 4.** Worm plots of residuals from the best non-stationary models for three distributions of annual groundwater storage variables, with *P*, *E*, and *n* as covariates. For a satisfactory fit, the data points should be within the two gray lines (95% confidence interval).

**Table 7.** Residual sequence statistical indicators and Filliben correlation coefficient of the best non-stationary model for three distributions of annual groundwater storage variables, with *P*, *E*, and *n* as covariates. $\nu$ represents the skewness coefficient, and $\tau$ refers to kurtosis coefficient.

| Distribution | Characteristics | Covariate Combination | $\mu$ | $\sigma$ | $\nu$ | $\tau$ | Filliben Coefficient |
|---|---|---|---|---|---|---|---|
| GU$-\mu$ | $\mu$ changing, $\sigma$ constant | $P + E + n$ | $-0.00$ | 1.00 | 0.11 | 2.42 | 0.99 |
| NO$-\sigma$ | $\mu$ constant, $\sigma$ changing | $P + n$ | $-0.17$ | 0.99 | 52 | 3.13 | 0.98 |
| GU$-\mu\sigma$ | $\mu$, $\sigma$ changing | $P + E + n$ | $-0.02$ | 1.02 | 60.8 | 2.37 | 0.99 |

4.1.4. Quantitative Contribution Estimation of *P*, *E*, and *n* to Groundwater Storage Changes

Statistical simulation methods were used to quantitatively examine the changes in groundwater storage variables compared to those at the baseline year of 1957, determining the contributions of precipitation, actual evapotranspiration, and underlying surface factors to these changes (Figure 5). Figure 5a indicates that precipitation and the underlying surface parameters positively contribute to changes in the groundwater storage variables, whereas actual evapotranspiration has a negative influence. Prior to the 1970s, precipitation, evapotranspiration, and the underlying surface characteristics in the Tuwei River Basin fluctuated dramatically. However, since 2000, the proportion of actual evapotranspiration and underlying surface parameters to groundwater storage variables has grown, especially

with evapotranspiration's contribution increasing rapidly and eventually surpassing that of the total precipitation and underlying surface parameters' contributions, causing the groundwater storage variable to shift from positive to negative over time. Figure 5b depicts a correlation scatter plot of the simulated groundwater storage variables and calculated values obtained by using the USGS RORA model [27]. The graph shows a correlation coefficient of 0.71 between the simulated groundwater storage variables and the calculated values obtained using the USGS RORA model, demonstrating that the inconsistent model created in this study performs well in this simulation.

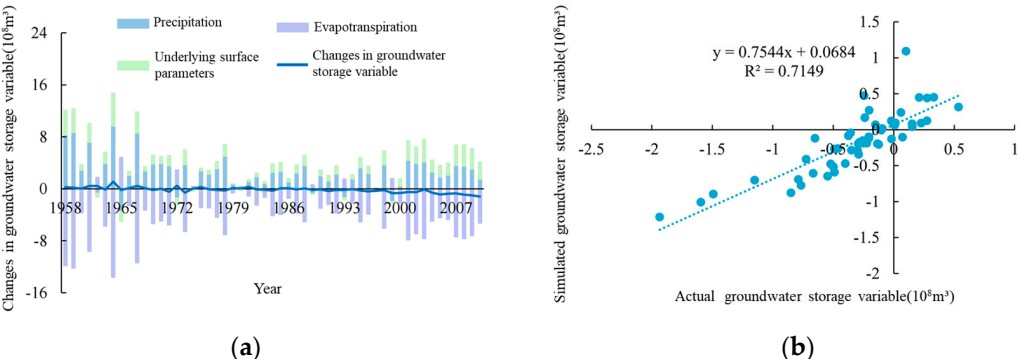

(a)                                                                                              (b)

**Figure 5.** Contribution changes (**a**) of *P*, *E*, and *n* to groundwater storage variables, and correlation scatter plot (**b**) between simulated and actual change values.

## 4.2. Statistical Attribution of Changes in Actual Evapotranspiration and Underlying Surface Variable

### 4.2.1. Actual Evapotranspiration

This study found that when the mean μ changes with the physical factors and the mean square error σ is constant, the GAMLSS of actual evapotranspiration with LO distribution has the lowest *AIC* value, with the yearly precipitation and annual underlying surface characteristics acting as covariates. The link between the mean *μ* and physical variables indicates that annual evapotranspiration rises with increasing precipitation and underlying surface variables. When the mean *μ* is constant and the mean square error *σ* varies with the physical factors, the GAMLSS of actual evapotranspiration with practically all the distributions has higher *AIC* values, in which annual precipitation and the underlying surface characteristics are used as covariates. The NO distribution has a relatively low *AIC* value. When both the mean *μ* and mean deviation *σ* change due to physical variables, the GAMLSS of evapotranspiration with LO distribution provides the lowest AIC value, implying that as annual precipitation increases, so does actual evapotranspiration, and it also grows as the underlying surface parameters increase. Figure 6 depicts the 5%, 50%, and 95% quantile plots of annual actual evapotranspiration fitted using the optimal non-stationary GAMLSS in the three cases above, as well as the residual sequence. Figure 7 and Table 8 illustrate a worm plot, statistical indicators, and the Filliben correlation coefficient.

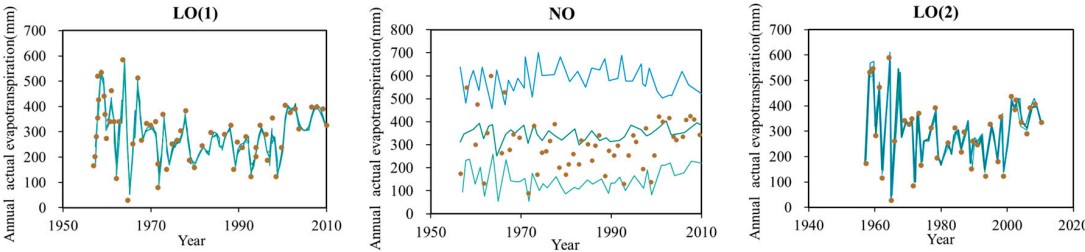

**Figure 6.** Five % (blue), fifty % (dark green), and ninety-five % (light green) quantile plots of the best non-stationary model for three distributions of actual evapotranspiration, with physical variables as covariates.

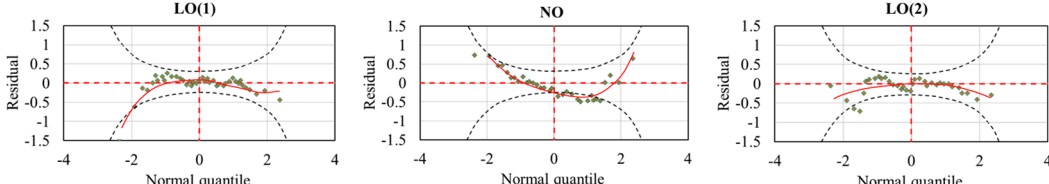

**Figure 7.** Worm plots of residuals from the best non-stationary models for three distributions of annual actual evapotranspiration, with *P* and *n* as covariates. For a satisfactory fit, the data points should be within the two gray lines (95% confidence interval).

**Table 8.** Residual sequence statistical indicators and Filliben correlation coefficient of the best non-stationary model for three distributions of annual actual evapotranspiration, with *P* and *n* as covariates. $\nu$ represents the skewness coefficient, and $\tau$ refers to kurtosis coefficient.

| Distribution | Characteristics | Covariate Combination | $\mu$ | $\sigma$ | $\nu$ | $\tau$ | Filliben Coefficient |
|---|---|---|---|---|---|---|---|
| LO$-\mu$ | $\mu$ changing, $\sigma$ constant | $P + n$ | −0.06 | 1.09 | −0.95 | 4.88 | 0.97 |
| NO$-\sigma$ | $\mu$ constant, $\sigma$ changing | $P$ | −0.18 | 0.71 | 1.63 | 6.27 | 0.92 |
| LO$-\mu\sigma$ | $\mu$, $\sigma$ changing | $P + n$ | −0.06 | 1.03 | −0.44 | 2.90 | 0.98 |

It can be seen in the chart that when the mean $\mu$ and mean deviation $\sigma$ change along with the physical factors, the LO distribution-based GAMLSS of annual actual evapotranspiration, with annual precipitation and the surface parameters as covariates, fits better when the physical factors change. It also has a smaller *AIC* value than both the consistent model and the inconsistent model with time as a covariate. The Filliben correlation coefficient exceeds 0.95, allowing it to better reflect the inconsistent fluctuations in annual actual evapotranspiration. According to Figure 8a, both precipitation and the underlying surface factors contribute positively to actual evapotranspiration variations. Precipitation and the underlying surface characteristics changed dramatically in the Tuwei River Basin prior to the 1970s. However, after 2000, the contribution of the underlying surface parameters to actual evapotranspiration increased, causing actual evapotranspiration to gradually climb. Figure 8b shows a scatter plot of the simulated values and calculated values obtained using a Budyko-based HWEB model [28]. This graphic demonstrates that the correlation coefficient between the two factors is as high as 0.99, showing that the non-stationary model has a good simulation effect.

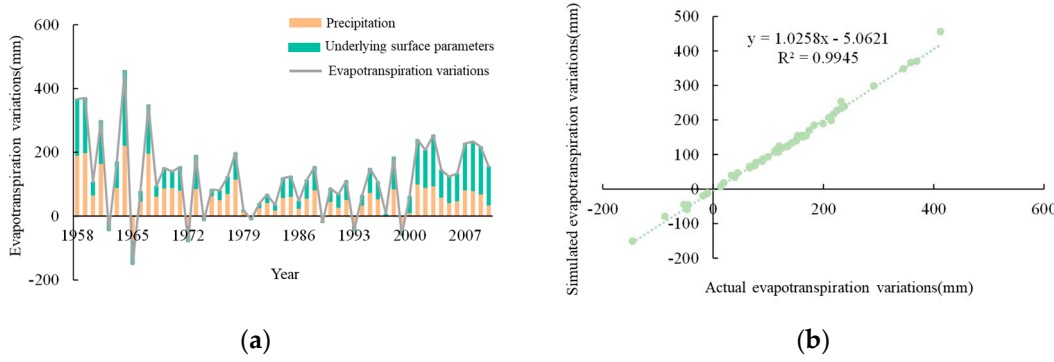

(**a**)  (**b**)

**Figure 8.** Contribution changes (**a**) in *P* and *n* to *E* and correlation scatter plot (**b**) between the changes in simulated and measured values.

### 4.2.2. Underlying Surface Variables

The GAMLSS of the underlying surface parameter *n,* with NO distribution and annual precipitation as covariates, has the smallest *AIC* value when the mean $\mu$ changes with the physical factors and the mean square error $\sigma$ remains constant. This suggests that as annual

precipitation increases, so do the underlying surface characteristics. When the mean $\mu$ is constant and the mean square error $\sigma$ varies with the physical parameters, the GAMLSSs with the underlying surface parameter $n$ with almost all the distributions have substantial *AIC* values, with the exception of the GA distribution, which has a relatively modest value. When both the mean $\mu$ and mean deviation $\sigma$ vary with the physical parameters, the GAMLSS with NO distribution has the lowest *AIC* value. The link between the mean $\mu$ and physical components indicates that the yearly underlying surface parameters fluctuate with precipitation. They increase as the quantity rises.

Figure 9 depicts the 5%, 50%, and 95% quantile plots of the yearly underlying surface parameters fitted using the optimally inconsistent GAMLSS, with precipitation as the covariate as in the previous three cases. Figure 10 and Table 9 show a residual sequence worm graph and statistical indicators, as well as the Filliben correlation coefficient.

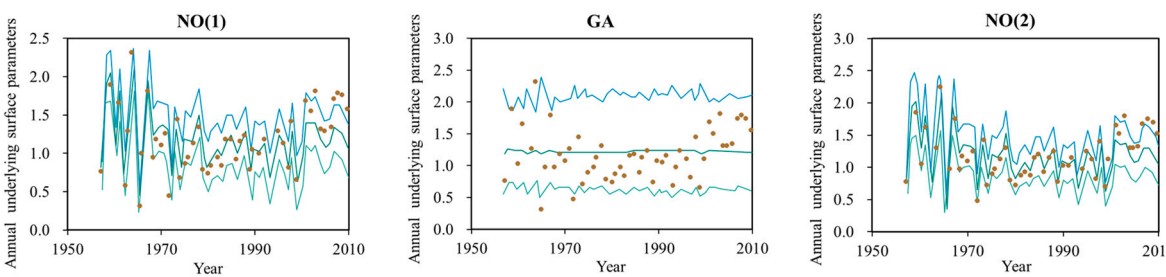

**Figure 9.** Five % (blue), fifty % (dark green), and ninety-five % (light green) quantile plots of the best non-stationary model for three distributions of underlying surface parameter, with *P* as the covariate.

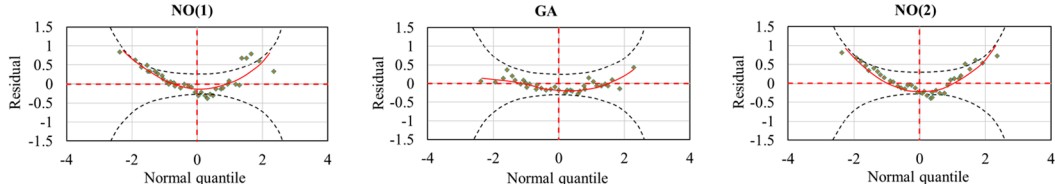

**Figure 10.** Worm plots of residuals from the best non-stationary models for three distributions of annual underlying surface parameter, with *P* as the covariate. For a satisfactory fit, the data points should be within the two gray lines (95% confidence interval).

**Table 9.** Residual sequence statistical indicators and Filliben correlation coefficient of the best non-stationary model for three distributions of annual underlying surface parameter, with *P* as the covariate.

| Distribution | Characteristics | Covariate Combination | $\mu$ | $\sigma$ | $\nu$ | $\tau$ | Filliben Coefficient |
|---|---|---|---|---|---|---|---|
| NO$-\mu$ | $\mu$ changing, $\sigma$ constant | *P* | 0.00 | 1.02 | 1.03 | 3.45 | 0.95 |
| GA$-\sigma$ | $\mu$ constant, $\sigma$ changing | *P* | $-0.11$ | 0.96 | 0.43 | 3.18 | 0.99 |
| NO$-\mu\sigma$ | $\mu$, $\sigma$ changing | *P* | $-0.00$ | 1.02 | 1.05 | 3.58 | 0.96 |

The graph indicates that the NO-GAMLSS using annual precipitation as a covariate, provides a better fit when the mean μ and mean deviation $\sigma$ change because of physical factors. Furthermore, its *AIC* value is lower than that of the consistent model with annual underlying surface parameters and the inconsistent model with time as a covariate. Furthermore, the Filliben correlation value exceeds 0.95, indicating that the model can better reflect non-consistent variations in the underlying surface parameters. According to Figure 11a, precipitation contributes positively to changes in the underlying surface characteristics. Precipitation's contribution fluctuated substantially up until the 1970s, which then decreased and stabilized. Figure 11b shows a scatter plot of the non-stationary model's simulated and calculated values from the HWEB model. As seen in the figure, the

correlation coefficient between the two factors is 0.76, indicating that the model provides better simulation results.

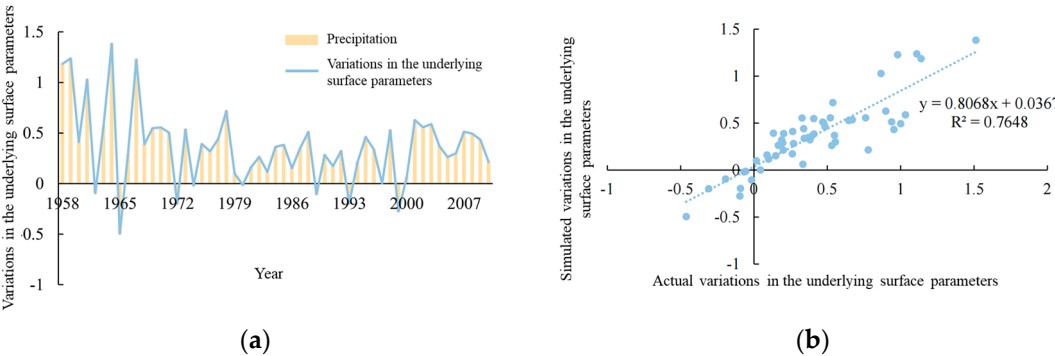

(**a**)  (**b**)

**Figure 11.** Contribution changes (**a**) of $P$ to underlying surface parameter $n$ and correlation scatter plot (**b**) between the changes in simulated and computed values.

### 4.3. Attribution of Changes in Runoff and Statistical Analysis Method Validation

Figure 2 and the basin's multi-year average water balance formula ($R = P - E + \Delta G$) were used to quantify the impact of simulated precipitation, actual evapotranspiration, and the groundwater storage variables on runoff. Figure 12 depicts the computation results, indicating that the changes in precipitation and groundwater storage variables, on average, contribute positively to runoff changes, whereas the changes in actual evapotranspiration contribute negatively. After 2000, the variations in actual evapotranspiration and groundwater storage variables had a stronger impact on runoff, resulting in a progressive reduction. Figure 12 depicts a scatter plot of observed runoff and the estimated one using the basin water balance method, with a correlation coefficient of 0.61, demonstrating that the simulation can effectively characterize the inconsistent changes in annual runoff.

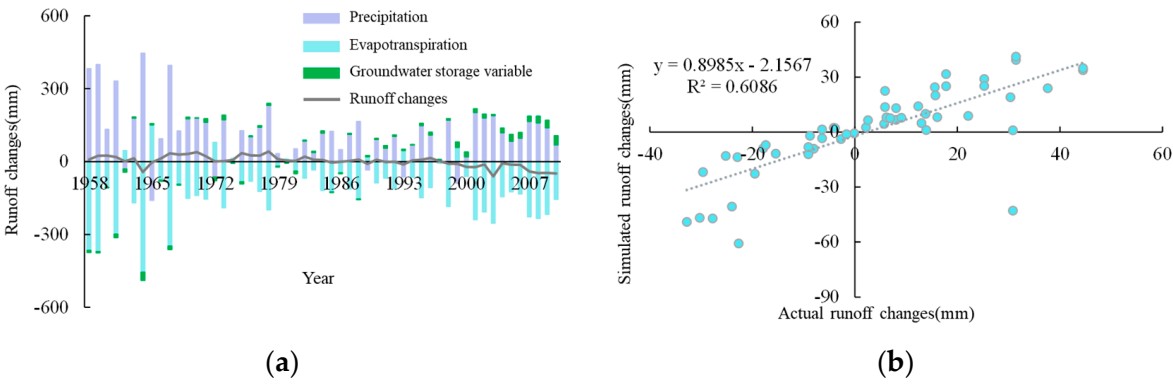

(**a**)  (**b**)

**Figure 12.** Contribution changes (**a**) of $P$, $E$, and $\Delta G$ to runoff and correlation scatter plot (**b**) between the changes in observed and computed values.

### 4.4. Scenario Simulation Using a Non-Stationary Model

#### 4.4.1. Baseline Conditions

Using multi-year average precipitation as the benchmark, we determined the changes in the underlying surface parameter $n$ caused by precipitation under the benchmark conditions, the actual evapotranspiration changes caused by precipitation and the underlying surface parameters, the changes in groundwater storage variables caused by surface characteristics and actual evapotranspiration, as well as the runoff changes caused by precipitation, actual evapotranspiration, and the groundwater storage variables. Table 10 shows the specific driving adjustments. The results show that the change in underlying surface parameters driven by precipitation (392.2 mm) is 1.17 in value. Precipitation and the underlying surface parameters drive an actual change in evapotranspiration of 298.31 mm.

The surface characteristics and actual evapotranspiration determine a groundwater storage variable of 5.75 mm, while precipitation, actual evapotranspiration, and the groundwater storage variables contribute to a runoff depth change of 88.14 mm.

**Table 10.** Calculations of variable changes under baseline conditions.

| Driving Process | Calculated Variables | Change Value |
|:---:|:---:|:---:|
| $P \rightarrow n$ | $n$ | 1.17 |
| $P \& n \rightarrow E$ | $E$ | 298.31 (mm) |
| $P, n \& E \rightarrow \Delta G$ | $\Delta G$ | 5.75 (mm) |
| $P, n \& \Delta G \rightarrow R$ | $R$ | 88.14 (mm) |

4.4.2. Precipitation Increasing by 10%

Assuming precipitation increases by 10% based on the multi-year average, the change in the underlying surface parameter $n$ driven by precipitation under this scenario, as well as the actual evapotranspiration change, are calculated sequentially. The runoff change values are determined using the calculated precipitation, actual evapotranspiration, and groundwater storage variables. Table 11 shows the estimated driving changes for various factors. The results demonstrate that the change value of the underlying surface parameters driven by the 10% increase in precipitation is 1.29 in value. The actual evapotranspiration change is 336.61 mm, which is caused by a 10% increase in precipitation and responsive changes in the underlying surface parameters. Further, the 10% increase in precipitation, underlying surface parameters, and actual evapotranspiration cause a change in groundwater storage value of 5.44 mm. Correspondingly, the difference in runoff depth is 89.37 mm. Comparing the values of each variable calculated under this scenario to the simulation results under baseline conditions reveals that a 10% increase in precipitation led to an value increase of 0.12 in the underlying surface parameters, an increase of 38.3 mm in actual evapotranspiration, a reduction of 0.31 mm in the groundwater storage variables, and an increase of 1.23 mm in the runoff depth.

**Table 11.** Calculations of variable changes under a 10% increase in precipitation.

| Driving Process | Calculated Variables | Change Value | Difference from Baseline Conditions |
|:---:|:---:|:---:|:---:|
| $P \rightarrow n$ | $n$ | 1.29 | +0.12 |
| $P \& n \rightarrow E$ | $E$ | 336.61 (mm) | +38.3 (mm) |
| $P, n \& E \rightarrow \Delta G$ | $\Delta G$ | 5.44 (mm) | −0.31 (mm) |
| $P, n \& \Delta G \rightarrow R$ | $R$ | 89.37 (mm) | +1.23 (mm) |

4.4.3. Underlying Surface Parameters Increasing by 10%

The multi-year average raises the underlying surface parameter $n$ by 10%. Next, the change in actual evapotranspiration caused by precipitation and the increased underlying surface parameter, the change in groundwater storage variables affected by precipitation, actual evapotranspiration, and increased underlying surface parameter, and runoff change values triggered by precipitation, actual evapotranspiration, and groundwater storage variables in this case, are calculated one after the other. Table 12 shows the changes that various variables have caused. The calculations reveal that precipitation causes a value change of 1.17 in the underlying surface parameters, while a 10% rise in the parameter results in a change of 1.29 in value. The actual evapotranspiration change value is 316.39 mm following the precipitation change, and the underlying surface parameter rise by 10%. The groundwater storage variable changes by 1.17 mm following changes in precipitation and actual evapotranspiration, as well as the underlying surface parameters rising by 10%. Furthermore, comparing the calculated change values of each variable in this scenario to the baseline conditions reveals that a 10% increase in the underlying surface parameters results in an 18.08 mm rise in evapotranspiration and a 6.32 mm decrease in groundwater storage. The depth of the flow dropped by 11.77 mm.

**Table 12.** Calculations of variable changes under a 10% increase in underlying surface parameters.

| Driving Process | Calculated Variables | Change Value | Difference from Baseline Conditions |
|:---:|:---:|:---:|:---:|
| $P \rightarrow n$ | $n$ | 1.17 | |
| $n$ increasing by 10% | Increase in $n$ | 0.12 | +0.12 |
| Total change of $n$ | $n$ | 1.29 | |
| $P$ & $n \rightarrow E$ | $E$ | 316.39 (mm) | +18.08 (mm) |
| $P$, $n$ & $E \rightarrow \Delta G$ | $\Delta G$ | −0.57 (mm) | −6.32 (mm) |
| $P$, $n$ & $\Delta G \rightarrow R$ | $R$ | 76.37 (mm) | −11.77 (mm) |

4.4.4. Both Precipitation and Surface Parameters Increasing by 10% Simultaneously

First, precipitation and the underlying surface parameters are both increased by 10% simultaneously. Next, a 10% increase in precipitation alters the underlying surface characteristics, followed by change in actual evapotranspiration by the two variables' increases. Furthermore, a 10% increase in precipitation, the underlying surface parameters, and actual evapotranspiration causes the quantity of groundwater storage variables to alter. Finally, the 10% increase in precipitation and actual evapotranspiration, as well as the groundwater storage variables causes the runoff depth value to shift. Table 13 shows the driving quantities for each variable. The table demonstrates that a 10% increase in precipitation alters the underlying surface parameter by 1.29 in value. The overall value of the underlying surface parameter increases by 10%, to 1.42. When precipitation increases by 10%, the underlying surface variables increase by 10%, actual evapotranspiration changes, and the groundwater storage variable decreases by 1.53 mm. When precipitation increases by 10% and actual evapotranspiration and the groundwater storage variables change proportionally, the depth of the flow driven increases by 76.38 mm. When the values for each variable in this scenario are compared to the baseline value, it is discovered that when both precipitation and underlying surface parameters rise by 10% at the same time, the underlying surface parameters rise by 0.25 in value, actual evapotranspiration rises by 58.26 mm, groundwater storage decreases by 7.28 mm, and runoff decreases by 11.76 mm.

**Table 13.** Calculations of variable changes under 10% increases in precipitation and underlying surface parameter simultaneously.

| Driving Process | Calculated Variables | Change Value | Difference from Baseline Conditions |
|:---:|:---:|:---:|:---:|
| $P \rightarrow n$ | $n$ | 1.29 | |
| $n$ increasing by 10% | Increase in $n$ | 0.13 | +0.25 |
| Total change of $n$ | $n$ | 1.42 | |
| $P$ & $n \rightarrow E$ | $E$ | 356.57 (mm) | +58.26 (mm) |
| $P$, $n$ & $E \rightarrow \Delta G$ | $\Delta G$ | −1.53 (mm) | −7.28 (mm) |
| $P$, $n$ & $\Delta G \rightarrow R$ | $R$ | 76.38 (mm) | −11.76 (mm) |

Looking at the simulation results of watershed runoff under the different scenarios shown above, the runoff goes up by 1.23 mm when precipitation goes up by 10%, down by 11.77 mm when the underlying surface parameters go up by 10%, and down by 11.76 mm when both precipitation and the underlying surface parameters go up by 10%. From this, it can be seen that the variables that cause flow have effects on each other, and these impacts change over time. Once their compounds and interaction patterns change, there will be complicated hydrological effects and changes in runoff that are nonlinear.

**5. Conclusions**

This study investigated the cross-driving relationship between watershed hydrometeorological (precipitation, actual evapotranspiration, groundwater storage variables, and runoff) and surface environmental variables (underlying surface parameters) with incon-

sistently changing characteristics by building a non-stationary GAMLSS for each driving variable with physical factors as covariates. The main findings are as follows:

The GU-distribution-based model with precipitation, actual evapotranspiration, and underlying surface parameters as covariates has a better fitting effect for the groundwater storage variables, with a smaller *AIC* value compared to those of the consistent model and inconsistent model with time as a covariate. After the year 2000, the patterns in time show that actual evapotranspiration and the underlying surface parameters in the basin became more important to the groundwater storage variables, while the provoked groundwater storage variables quickly decreased.

The LO-distribution-based model, which includes precipitation and the underlying surface parameters as covariates, has a superior match for actual evapotranspiration, with an *AIC* value that is lower than those of the consistent model and inconsistent model, which included time as a covariate. After 2000, the contribution of underlying surface parameters to the actual evapotranspiration significantly increased, leading to an increase in evapotranspiration in the basin.

For the yearly underlying surface parameters, the NO-distribution-based model with precipitation as a covariate fits better when both the mean $\mu$ and the mean deviation $\sigma$ change because of physical factors. Additionally, its *AIC* value is lower than that of the consistent sexual model and the non-uniform model with time as a covariate.

The findings in interacting effects revealed that precipitation and the groundwater storage variables had positive contributions to runoff variations, whereas actual evapotranspiration had a negative effect. Particularly, from 2000, variations in actual evapotranspiration and the groundwater storage variables continued to contribute to the runoff changes year after year, resulting in a progressive decrease in runoff.

The scenario simulation results demonstrate that if precipitation increases by 10%, the underlying surface parameters increase the value by 0.12, actual evapotranspiration increases by 38.3 mm, the groundwater storage variable reduces by 0.31 mm, and runoff increases by 1.23 mm. In the situation where the underlying surface characteristics increase by 10%, actual evapotranspiration rises by 18.08 mm, groundwater storage falls by 6.32 mm, and runoff falls by 11.77 mm. In the scenario where precipitation and underlying surface parameters both increase by 10%, the underlying surface parameters increase the value by 0.25, actual evapotranspiration increases by 58.26 mm, groundwater storage variable decreases by 7.28 mm, and runoff decreases by 11.76 mm. It is clear that there are cross-influences across the driving variables for runoff, and this influence has time-varying properties, resulting in more complex hydrological impacts and a nonlinear runoff change.

**Author Contributions:** Conceptualization, H.Z.; methodology, T.Z.; software, C.D.; validation, S.Z., T.Z. and C.Y.; formal analysis, S.Z.; investigation, D.M. and Y.Z.; resources, F.L.; data curation, S.L.; writing—original draft preparation, S.Z.; writing—review and editing, H.Z.; visualization, T.Z.; supervision, H.Z.; project administration, H.Z.; funding acquisition, H.Z. All authors have read and agreed to the published version of the manuscript.

**Funding:** This research was funded by National Natural Science Foundation of China, grant numbers 51979005 and 52379003, Natural Science Basic Research Program of Shaanxi, grant number 2022JC-LHJJ-03, and Special Fund for Basic Research Funds of Central Universities, grant number 300102293201.

**Data Availability Statement:** The data presented in this study are available on request from the corresponding author.

**Acknowledgments:** Our cordial thanks are extended to the editor and anonymous reviewers for their pertinent and professional suggestions and comments which are greatly helpful for the further improvement of the quality of this paper.

**Conflicts of Interest:** The authors declare no conflicts of interest.

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
