# Peer review of "Statistical Analysis and Scenario Modeling of Non-Stationary Runoff Change in the Loess Plateau: A Novel Application of the Generalized Additive Model in Location, Scale and Shape"

_water, doi:10.3390/w16070986_

Round 1
Reviewer 1 Report
Comments and Suggestions for Authors
The manuscript analyzes the cross-driving relationship between runoff changes and its driving variables, taking as a case study Tuwei River Basin, first-level tributary of the Yellow River, Shaanxi Province, China.
The paper develop an inconsistent hydrological statistical model with physical factors as covariates to identify the watershed hydrometeorological and surface environmental variables with inconsistent changing characteristics.
I have the following Major observations:
1. The Abstract section is too large. Try to compress it and keep the most important information;
2. Regarding the presented Methodology:
- its structure (stages) are well defined and correct. However, the new contributions brought by the authors and the manuscript are not presented clearly enough. I recommend the introduction of a centralizing table in the Introduction Section, in which to present the new elements in a synthesized way.
- there is no rigorous justification (statistical, mathematical and hydrological) regarding the choice of statistical distributions analyzed and used in the manuscript. Seven statistical distributions are analyzed, with different number of parameters, which is not recommended using the maximum likelihood method, because having different numbers of parameters, the distributions calibrate different upper moments (the same situation generally occurs when using the method of ordinary moments). It is recommended to use only three-parameter distributions, or only two-parameter distributions. The selection criteria can constitute a rigorous criterion only in the field of registered values.
- it is used as a parameter estimation method, the maximum likelihood method, being one of the most deficient parameter estimation methods, especially in the case of short and medium lengths of observed data (records). It is recommended to use the L-moments method, the most stable and robust, being the least influenced by small lengths (<30 values) and medium lengths of recorded data (<50 values). There are many articles published in 2023 in Water, which demonstrated this.
3. Regarding the documentation, it is too brief (only 21 references). The disadvantages and advantages of the methods used are not presented. No new contributions are mentioned, from the last period, regarding the analyzed distributions (In Water, in 2023, many materials were published regarding these elements.
Author Response
Dear Reviewer:
Thank you for your feedback on our paper, " Statistical analysis and scenario modeling of non-stationary runoff change in the Loess Plateau: A novel application of GAMLSS model (water-2922860)". Those are excellent and helpful comments. We attentively read over the comments and responded accordingly. We uploaded the revised manuscript file following the directions in your letter. Text revisions are highlighted in red for modification. The following are the responses to the reviewer's comments.
We genuinely appreciate your time and consideration on this revised version of the paper.
**********************************************************************************************************
The manuscript analyzes the cross-driving relationship between runoff changes and its driving variables, taking as a case study Tuwei River Basin, first-level tributary of the Yellow River, Shaanxi Province, China.
The paper develop an inconsistent hydrological statistical model with physical factors as covariates to identify the watershed hydrometeorological and surface environmental variables with inconsistent changing characteristics.
I have the following Major observations:
- The Abstract section is too large. Try to compress it and keep the most important information;
Response: Thank you for your valuable comment. We have condensed the manuscript's abstract to just include the essential details per your suggestion. The revised Abstract is as follows.
Abstract: The hydrological series in the Loess Plateau region has exhibited a shift in trend, mean, and/or variance as environmental conditions have changed, indicating a departure from the assumption of stationarity. As the variations accumulate, the compound effects caused by the driving variables on runoff variations grow complex and interactive, posing a substantial risk to water security and the promotion of high-quality development in regions or river basins. This study focuses on the Tuwei River Basin in the Loess Plateau, which experiences significant changes in vegetation coverage and low human disturbance, and examines the cross-driving relationship between runoff changes and its driving variables (including hydrometeorological and environmental variables). A quantitative statistical analysis method based on GAMLSS is then developed to estimate the interacting effects of changes in driving variables and their contribution to runoff changes. Finally, various anticipated scenarios are used to simulate changes in driving variables and runoff disturbances. The findings indicate: (1) The developed GU, LO, and NO distribution-based GAMLSS models provide a notable advantage in effectively capturing variations in groundwater storage variables, actual evapotranspiration, and underlying surface parameters, as well as accurately estimating the impacts of other relevant variables. (2) Precipitation and groundwater storage variables showed predominantly positive contributions to runoff change, but actual evapotranspiration had an adverse effect. Changes in underlying surface parameters, particularly since 2000, increase actual evapotranspiration while decreasing groundwater storage variables, resulting in a progressive decrease in runoff as their contribution grows. (3) The scenario simulation results reveal that alterations to the underlying surface have a substantial influence on the evolution of runoff in the Tuwei River Basin. Additionally, there are cross-effects between the impact of various driving variables on runoff, potentially compounding the complexity of inconsistent changes in runoff sequences.
- Regarding the presented Methodology:
- its structure (stages) are well defined and correct. However, the new contributions brought by the authors and the manuscript are not presented clearly enough. I recommend the introduction of a centralizing table in the Introduction Section, in which to present the new elements in a synthesized way.
Response: Thank you for your helpful feedback. We rewrote the Introduction section and emphasized the new contributions brought by the authors.
- Introduction
Precipitation, evaporation, water storage, and the underlying surface condition are commonly regarded as the primary causes of watershed runoff. Recently, they have seen varying degrees of change, impacted by increasingly intense climate change and human activities around the world, primarily generating non-stationary changes in runoff in arid and semi-arid regions [1-2]. Furthermore, the elements impact and push each other, amplifying their interactions and driving effects on runoff in these regions [3-5]. The Loess Plateau (LP) is the most common area, and it is receiving increased attention, particularly since the vegetation cover has altered dramatically [6-8]. As widely described in the literature, major elements in the LP's water cycle process have changed dramatically in recent years as a result of climate change and human-induced vegetation regreening [9]. Runoff, as the result of changes in numerous elements, is heavily influenced by the compound impacts of these changes [7, 10-11]. Physically, precipitation, evapotranspiration, and groundwater storage have a direct impact on runoff change, whereas changes in the underlying surface are the primary environmental factors that indirectly cause runoff changes. However, when the streamflow is non-stationary, it is extremely difficult to disentangle the contributions of various elements to the inconsistent change in runoff to figure out the attribution explanations of significant variations in runoff on the LP.
As a result, many scientists have explored the inconsistency and attribution of regional hydrological processes using various time series models or physical hydrological models. For example, Feng et al. (2016) used a multiple regression model to investigate the interactive roles of climate and human activities on runoff decline in 14 basins in the LP, and discovered that reduced precipitation was the primary reason for the decrease in runoff between 1961 and 2009, with human intervention playing a dominant role in producing the shift points after which water yield decreased further [6]. Zhang et al. (2020) used the partial least squares regression (PLSR) approach to evaluate the contribution of the expanding implementation of ecological restoration (ER) strategies in the LP to streamflow decline, and the results revealed that ER was the dominant cause of streamflow reduction, with a contribution increasing from 59% in 1980-1999 to 82% in 2000-2015 [10]. Tan et al. (2024) proposed a modified Budyko attribution method to quantify vegetation-induced runoff alterations in the LP, and the findings show that the vegetation change mainly caused runoff reduction over the LP, resulting in 78.94% of the reduced runoff, and the "Grain-for-Green" Program (GFGP)-led LULC shift, particularly cropland reduction, plays a vital role in vegetation-induced runoff losses, which could increase future water stress in the LP. Gao et al. (2019) used the SWAT model to simulate run-off change under several scenarios in the Jing River Basin of the LP, and evaluated the climatic and anthropogenic impacts. The results showed that the impact of climatic elements was progressively diminishing over time, while the influence of direct variables (water withdrawal) was expanding the fastest, and the influence of indirect causes was steadily increasing. Sun et al. (2019) used the RCC-WBM model to quantitatively separate the impacts of climate change and human activities on runoff change in the Tao River from the Tibetan Plateau to the LP, and found that human activities are the primary drivers of runoff reduction in the Basin, though both absolute influences tend to increase [13]. Liu et al. (2012) used the Tsinghua Hydrological model based on the Representative Elementary Watersheds approach (THREW) to investigate the characteristics of runoff generation in the LP, concluding that the subsurface flow contribution to total streamflow is greater than 53% from October to March, while the overland flow contribution exceeds 72% from April to September [14].
Although the aforementioned research has made significant progress in interpreting the interactive effects of the hydrological cycle on the LP, as well as analyzing the attribution of non-stationary characteristics in runoff sequences, these two models or methods have some limitations. For example, the time series analysis can only provide average and general contribution estimates, such as from climate change, ecological restoration (ER) strategies, or human activities. In other words, while these methods may be beneficial for analyzing nonstationary time series, their intrinsic static regression aspect does not properly describe many complicated physical interaction processes [15]. Hydrological models, such as lumped (IHACRES), semi-distributed (HEC-HMS), and fully-distributed (SWATgrid), accurately simulate runoff in smaller catchments under most hydroclimatic conditions, but they frequently fail in larger catchments, regardless of hydroclimatic conditions [16], particularly in catchments with non-stationarity in rainfall-runoff relationships. This implies a need for additional research to promote the establishment of the cross-driving relationship between runoff non-stationary changes and their driving elements, as well as the quantitatively evaluation of variations in the driving elements and their impact on runoff alterations, both of which are critical in formulating water resource management policy and addressing water shortages on the LP.
The Generalized Additive Model in Location, Scale and Shape (GAMLSS) model is a tool for modeling time series under non-stationary conditions [17]. It supports a variety of random variable frequency distribution types, and is extremely useful in constructing linear or nonlinear functional relationships between distribution function position parameters, scale parameters, shape parameters, and explanatory variables [18]. The GAMLSS framework has been widely applied in non-stationary frequency analysis, modeling, and forecasting in hydrology [19-22]. This GAMLSS feature also allows for cross-driving interactions between runoff and driving elements, or between driving elements themselves. In light of this, the goal of this paper is to create an inconsistent hydrological statistical model (GAMLSS) with physical factors as covariates in order to identify the cross-driving relationship between watershed hydrometeorological (precipitation, actual evapotranspiration, groundwater storage variables, and runoff) and surface environmental variables (underlying surface parameters) with inconsistently changing characteristics. It offers a novel approach to inquiry that differs from previously time series analysis and hydrological models in that it statistically analyzes the cross-driving relationship between elements and investigates the causes of runoff decline.
The study entails (1) investigating the cross-driving relationship between runoff changes and their driving delivers (including hydrometeorological and surface environmental elements), (2) developing a hydrological non-stationary model that incorporates several driving elements, (3) quantitatively examining changes in runoff driving variables and their impact on runoff evolution, and (4) developing scenario plans to simulate future runoff change patterns under various driving impacts. Although the study focuses on the Loess Plateau, the findings are extremely relevant to water managers in other arid and semi-arid regions with substantial hydroclimatic fluctuation or change.
Reference:
- Deb, P., Kiem, A. S., & Willgoose, G. (2019). Mechanisms influencing non-stationarity in rainfall-runoff relationships in southeast Australia. Journal of Hydrology, 571, 749-764. https://doi.org/10.1016/j.jhydrol.2019.02.025.
- Zhang, Q., Gu, X., Singh, V. P., Xiao, M., & Chen, X. (2015). Evaluation of flood frequency under non-stationarity resulting from climate indices and reservoir indices in the East River basin, China. Journal of Hydrology, 527, 565-575. https://doi.org/10.1080/02626667.2020.1754420.
- Deb, P., Kiem, A. S., & Willgoose, G. (2019). A linked surface water-groundwater modelling approach to more realistically simulate rainfall-runoff non-stationarity in semi-arid regions. Journal of Hydrology, 575, 273-291.. https://doi.org/10.1016/j.jhydrol.2019.05.039.
- Kling, H., Stanzel, P., Fuchs, M., & Nachtnebel, H. P. (2015). Performance of the COSERO precipitation–runoff model under non-stationary conditions in basins with different climates. Hydrological sciences journal, 60(7-8), 1374-1393. https://doi.org/10.1080/02626667.2014.959956.
- Ajami, H., Sharma, A., Band, L. E., Evans, J. P., Tuteja, N. K., Amirthanathan, G. E., & Bari, M. A. (2017). On the non-stationarity of hydrological response in anthropogenically unaffected catchments: an Australian perspective. Hydrology and Earth System Sciences, 21(1), 281-294. https://doi.org/10.5194/hess-21-281-2017.
- Feng, X., Cheng, W., Fu, B., & Lü, Y. (2016). The role of climatic and anthropogenic stresses on long-term runoff reduction from the Loess Plateau, China. Science of the Total Environment, 571, 688-698. https://doi.org/10.1016/j.scitotenv.2016.07.038.
- Tan, X., Jia, Y., Yang, D., Niu, C., & Hao, C. (2024). Impact ways and their contributions to vegetation-induced runoff changes in the Loess Plateau. Journal of Hydrology: Regional Studies, 51, 101630. https://doi.org/10.1016/j.ejrh.2023.101630.
- Yao, C., Zhang, H., Zhang, S., Dang, C., Mu, D., Yu, Z., & Lyu, F. (2024). A categorical quantification of the effects of vegetation restorations on streamflow variations in the Loess Plateau, China. Journal of Hydrology, 628, 130577. https://doi.org/10.1016/j.jhydrol.2023.130577.
- Li, Y., Mao, D., Feng, A., & Schillerberg, T. (2019). Will human-induced vegetation regreening continually decrease runoff in the loess plateau of China?. Forests, 10(10), 906. https://doi.org/10.3390/f10100906.
- Zhang, J., Gao, G., Li, Z., Fu, B., & Gupta, H. V. (2020). Identification of climate variables dominating streamflow generation and quantification of streamflow decline in the Loess Plateau, China. Science of the Total Environment, 722, 137935. https://doi.org/10.1016/j.scitotenv. 2020.137935.
- Gao, Z., Zhang, L., Cheng, L., Zhang, X., Cowan, T., Cai, W., & Brutsaert, W. (2015). Groundwater storage trends in the Loess Plateau of China estimated from streamflow records. Journal of Hydrology, 530, 281-290. https://doi.org/10.1016/j.jhydrol.2015.09.063.
- Gao, X., Yan, C., Wang, Y., Zhao, X., Zhao, Y., Sun, M., & Peng, S. (2020). Attribution analysis of climatic and multiple anthropogenic causes of runoff change in the Loess Plateau—A case‐study of the Jing River Basin. Land Degradation & Development, 31(13), 1622-1640. https://doi.org/1002/ldr.3557.
- Sun, L., Wang, Y. Y., Zhang, J. Y., Yang, Q. L., Bao, Z. X., Guan, X. X., Guan T. S., Chen X., & Wang, G. Q. (2019). Impact of environmental change on runoff in a transitional basin: Tao River Basin from the Tibetan Plateau to the Loess Plateau, China. Advances in Climate Change Research, 10(4), 214-224. https://doi.org/10.1016/j.accre.2020.02.002.
- Liu, D., Tian, F., Hu, H., & Hu, H. (2012). The role of run-on for overland flow and the characteristics of runoff generation in the Loess Plateau, China. Hydrological Sciences Journal, 57(6), 1107-1117. https://doi.org/10.1080/02626667.2012.695870.
- Coulibaly, P., & Baldwin, C. K. (2005). Nonstationary hydrological time series forecasting using nonlinear dynamic methods. Journal of Hydrology, 307(1-4), 164-174.https://doi.org/10.1016/ jhydrol.2004.10.008.
- Deb, P., & Kiem, A. S. (2020). Evaluation of rainfall–runoff model performance under non-stationary hydroclimatic conditions. Hydrological Sciences Journal, 65(10), 1667-1684. https://doi.org/10.1080/02626667.2020.1754420.
- Rigby, R. A., & Stasinopoulos, D. M. (2005). Generalized additive models for location, scale and shape. Journal of the Royal Statistical Society Series C: Applied Statistics, 54(3), 507-554.https://doi.org/10.1111/j.1467-9876.2005.00510.x.
- Serinaldi, F. (2011). Distributional modeling and short-term forecasting of electricity prices by generalized additive models for location, scale and shape. Energy Economics, 33(6), 1216-1226.https://doi.org/10.1016/j.eneco.2011.05.001.
- Scala, P., Cipolla, G., Treppiedi, D., & Noto, L. V. (2022). The Use of GAMLSS Framework for a Non-Stationary Frequency Analysis of Annual Runoff Data over a Mediterranean Area. Water, 14(18), 2848.https://doi.org/10.3390/w14182848.
- He, C., Chen, F., Long, A., Luo, C., & Qiao, C. (2021). Frequency Analysis of Snowmelt Flood Based on GAMLSS Model in Manas River Basin, China. Water, 13(15), 2007.https://doi.org/10.3390/
- Shiau, J. T., & Liu, Y. T. (2021). Nonstationary analyses of the maximum and minimum streamflow in Tamsui River basin, Taiwan. Water, 13(6), 762.https://doi.org/10.3390/w13060762.
- Rashid, M. M., & Beecham, S. (2019). Simulation of streamflow with statistically downscaled daily rainfall using a hybrid of wavelet and GAMLSS models. Hydrological sciences journal, 64(11), 1327-1339.https://doi.org/10.1080/02626667.2019.1630742.
- there is no rigorous justification (statistical, mathematical and hydrological) regarding the choice of statistical distributions analyzed and used in the manuscript. Seven statistical distributions are analyzed, with different number of parameters, which is not recommended using the maximum likelihood method, because having different numbers of parameters, the distributions calibrate different upper moments (the same situation generally occurs when using the method of ordinary moments). It is recommended to use only three-parameter distributions, or only two-parameter distributions. The selection criteria can constitute a rigorous criterion only in the field of registered values.
Response: Thank you for your significant comments. We completely agree with your statement.
According to your comments, we remove the only three-parameter distribution, generalized gamma/GG, and keep six two-parameter distributions: Gumbel/GU, normal/NO, logistic/LO, gamma/GA, log nor-mal/LOGNO, and Weibull/WEI.
In addition, we provide explanation for picking the statistical distributions investigated and used in the study.
Initially, a consistency analysis of each variable sequence in this study was carried out. Six two-parameter distributions were chosen: Gumbel/GU, normal/NO, logistic/LO, gamma/GA, log normal/LOGNO, and Weibull/WEI, which are commonly employed in frequency analysis of extreme events around the world [30-32], particularly in north China [33-34]. Their fit was tested in the GAMLSS model by fixing the stable model's parameters, with the goal of determining the best appropriate consistency probability distributions for each sequence. The three most suitable distributions of each sequence were then submitted to a non-consistency analysis with time as a covariate, and the parameters in the GAMLSS model were adjusted over time to examine the trend of each sequence parameter changing over time. The sequence is judged to have undergone non-consistent changes if the AIC and SBC values of the GAMLSS model with time as a covariate are less than the consistency model. Finally, the non-consistent GAMLSS model of runoff with physical factors as covariates is used to investigate non-consistent variations in runoff due to the influence of physical driving variables.
Reference:
- Scala, P.; Cipolla, G.; Treppiedi, D.; Noto, L.V. (2022). The Use of GAMLSS Framework for a Non-Stationary Frequency Analysis of Annual Runoff Data over a Mediterranean Area. Water, 14, 2848. https://doi.org/10.3390/w14182848.
- Eljabri, S. (2013). New statistical models for extreme values. The University of Manchester (United Kingdom).
- Ekwueme, B. N., Ibeje, A. O., & Ekeleme, A. (2021). Modelling of maximum annual flood for regional watersheds using markov model. Saudi Journal of Civil Engineering, 5(2), 26-34. https://doi.org/10.36348/sjce.2021.v05i02.002.
- Li, Z., Li, Z. Zhao, W., & Wang, Y. (2015). Probability Modeling of Precipitation Extremes over Two River Basins in Northwest of China. Advances in Meteorology, 2015, 374127. https://doi.org/10.1155/2015/374127.
- Li, L., Yao, N., Li Liu, D., Song, S., Lin, H., Chen, X., & Li, Y. (2019). Historical and future projected frequency of extreme precipitation indicators using the optimized cumulative distribution functions in China. Journal of hydrology, 579, 124170. https://doi.org/10.1016/j.jhydrol.2019.124170.
- it is used as a parameter estimation method, the maximum likelihood method, being one of the most deficient parameter estimation methods, especially in the case of short and medium lengths of observed data (records). It is recommended to use the L-moments method, the most stable and robust, being the least influenced by small lengths (<30 values) and medium lengths of recorded data (<50 values). There are many articles published in 2023 in Water, which demonstrated this.
Response: We appreciate your constructive feedback. We employed the L-moments approach for parameter estimation and found that the results were similar to those obtained using the maximum likelihood method, with a deviation of less than 5.6%. Thus, we kept the original results. However, we strongly agree with your opinion that the L-moments method is recommended for use when the length of collected data is shorter than 50. As a result, we include an additional comment in the Methodology section to urge readers to use suitable parameter estimation methods.
where yt is the measured value of the sequence, n is its length or the sample size, and F is the cumulative probability distribution function that the series follows. However, it should be noted that the L-moments method is recommended for use when the length of collected data is shorter than 50.
- Regarding the documentation, it is too brief (only 21 references). The disadvantages and advantages of the methods used are not presented. No new contributions are mentioned, from the last period, regarding the analyzed distributions (In Water, in 2023, many materials were published regarding these elements.
Response: Thank you for your valuable feedback, which will help us enhance our manuscript significantly.
Because this work focuses primarily on a new application of GAMLSS for modeling the interaction of runoff and its driving elements, it makes few recommendations for GAMLSS improvement. However, we have included some of GAMLSS's most recent research achievements in the references to enable readers stay up to date on the current research developments.
Introduction
The Generalized Additive Model in Location, Scale and Shape (GAMLSS) model is a tool for modeling time series under non-stationary conditions [17]. It supports a variety of random variable frequency distribution types, and is extremely useful in constructing linear or nonlinear functional relationships between distribution function position parameters, scale parameters, shape parameters, and explanatory variables [18]. The GAMLSS framework has been widely applied in non-stationary frequency analysis, modeling, and forecasting in hydrology [19-22]. This GAMLSS feature also allows for cross-driving interactions between runoff and driving elements, or between driving elements themselves. In light of this, the goal of this paper is to create an inconsistent hydrological statistical model (GAMLSS) with physical factors as covariates in order to identify the cross-driving relationship between watershed hydrometeorological (precipitation, actual evapotranspiration, groundwater storage variables, and runoff) and surface environmental variables (underlying surface parameters) with inconsistently changing characteristics. It offers a novel approach to inquiry that differs from previously time series analysis and hydrological models in that it statistically analyzes the cross-driving relationship between elements and investigates the causes of runoff decline.
References
- Rigby, R. A., & Stasinopoulos, D. M. (2005). Generalized additive models for location, scale and shape. Journal of the Royal Statistical Society Series C: Applied Statistics, 54(3), 507-554.https://doi.org/10.1111/j.1467-9876.2005.00510.x.
- Serinaldi, F. (2011). Distributional modeling and short-term forecasting of electricity prices by generalized additive models for location, scale and shape. Energy Economics, 33(6), 1216-1226.https://doi.org/10.1016/j.eneco.2011.05.001.
- Scala, P., Cipolla, G., Treppiedi, D., & Noto, L. V. (2022). The Use of GAMLSS Framework for a Non-Stationary Frequency Analysis of Annual Runoff Data over a Mediterranean Area. Water, 14(18), 2848.https://doi.org/10.3390/w14182848.
- He, C., Chen, F., Long, A., Luo, C., & Qiao, C. (2021). Frequency Analysis of Snowmelt Flood Based on GAMLSS Model in Manas River Basin, China. Water, 13(15), 2007.https://doi.org/10.3390/
- Shiau, J. T., & Liu, Y. T. (2021). Nonstationary analyses of the maximum and minimum streamflow in Tamsui River basin, Taiwan. Water, 13(6), 762.https://doi.org/10.3390/w13060762.
- Rashid, M. M., & Beecham, S. (2019). Simulation of streamflow with statistically downscaled daily rainfall using a hybrid of wavelet and GAMLSS models. Hydrological sciences journal, 64(11), 1327-1339.https://doi.org/10.1080/02626667.2019.1630742.
3.2 GAMLSS model
Generalized Additive Models in Location, Scale, and Shape (GAMLSS) is a semi-parametric regression model that analyzes the frequencies of stationary and non-stationary runoff and other features [17-22, 29-30].
References
- Shao, S., Zhang, H., Singh, V. P., Ding, H., Zhang, J., & Wu, Y. (2022). Nonstationary analysis of hydrological drought index in a coupled human-water system: Application of the GAMLSS with meteorological and anthropogenic covariates in the Wuding River basin, China. Journal of Hydrology, 608, 127692. https://doi.org/10.1016/j.jhydrol.2022.127692.
- Scala, P.; Cipolla, G.; Treppiedi, D.; Noto, L.V. (2022). The Use of GAMLSS Framework for a Non-Stationary Frequency Analysis of Annual Runoff Data over a Mediterranean Area. Water, 14, 2848. https://doi.org/10.3390/w14182848.
In addition, we revised much of the content and included more references (the total number of references now stands at 36) to provide more chances for extending or tracking reading.
Thank you very much for your consideration and comment on our manuscript.

Reviewer 2 Report
Comments and Suggestions for Authors
L55: That "as is well known..." for a statement on national geography...
the authors might as well replace such a sentence with: "as
demonstrated by X et al, and Y et al"... /major factors in the water
cycle process on the Loess Plateau have changed significantly as a
result of climate change and human activity, and runoff, as the end
product of changes in many elements, is greatly influenced by the
compound effects of these changes. Since this is a scientific article
and not a lecture, the authors should base their assertions on
scientific results which, in all cases, entail a certain "uncertainty".
I therefore suggest to the authors that, from now on, they should
moderate their claims on the basis of sound scientific publications.
L91: It seems to me that this is the first time that the abbreviation GAMLSS appears here, with the exception of the abstract. Would it be a good idea for the authors, before using the abbreviation, to write down its full meaning? As they have done in previous lines with other abbreviations.
L99: I still see a certain "sufficiency" on the part of the authors which, for a scientific publication, seems unethical to me.
L104-107: It is confusing. Could you rephrase it so that the reader does not have to reread it to understand everything?
L148: Figure 1 has little graphical definition.
L152-154: This is what the authors claim... Where is it proven, and will they prove it later in this article? Is a bibliographic citation necessary to support such a claim? Check it out!
L154-159: So what now? Now they quote two articles and refer to what they stated in the previous sentences? This way it is not possible to understand what the authors mean. This needs to be rephrased, otherwise readers will not know what they are referring to.
This whole paragraph at the beginning of the Methodology section is confusing and repetitive of what the authors have said in previous paragraphs. I suggest rephrasing it so that it serves as an "introduction" to "methodology", but expressed in an orderly and simple way.
L168-184: This whole section is very interesting and relevant. But... Has it been fully developed by the authors, and has no one in the world, or before, contributed anything about it?
L188-192: I do not think that the quote provided by the authors is the
most appropriate for the technique they are describing. Through a simple
web search, any reader can see that, in a much more explicit and
extensive way, the skill on GAMLSS can be obtained after reading the
more related references.
Therefore, I suggest to the authors to be more scientifically ethical and instead of citing only articles of their nationality and interest, cite those who have developed the specific techniques. In addition, they should think of the readers, whose scientific ambition perhaps exceeds the topic the authors themselves are pursuing in this article.
L227-232: Once again... the values provided by the authors as a reference on the goodness of fit of the model, or the goodness of fit of the model... where do they come from? Is it the authors who determine this validity?
L338: Figure 3 has such a low definition that it is very difficult to understand it.
L350: Figure 4 has such a low definition that it is very difficult to understand it.
L374: Figure 5 has such a low definition that it is very difficult to understand it.
L397: Figure 6 has such a low definition that it is very difficult to understand it.
L399: Figure 7 has such a low definition that it is very difficult to understand it.
With the rest of the figures it is more or less the same: the definition is so low that you can hardly read the titles of the axes, the levels and the graphs themselves. Anyway...
L553-563: Up to the last sentence, the authors summarise in the conclusion section. Then the conclusions begin.
Well, I propose to remove all that summary (which is already explained above, and summarised in the abstract itself) and start with the conclusions, after all, this is "their section".
L565: “The XX-distribution-based GAMLSS model”. We already know very well that it is based on the GAMLSS model, perhaps it is not necessary to repeat it for all the models analyzed.
L578-581: It is difficult to understand. I recommend writing it in a simpler way. Furthermore, I suppose that the authors have slipped in a surprising error in the translation: "the AIC value is less than a consistent sexual model"; I suggest you be more careful when translating your scientific interpretation of your results.
L582-583: Readers already know that. They should reformulate the paragraph so that this, which is repeated in previous paragraphs of the article, is integrated and can be read with the meaning of what the authors wish to conclude next.
L589-599: It would seem to the authors to be a good idea to add the percentage increase of the different variables of interest. That is, "if precipitation increases by 10%, the underlying surface parameters increase by 0.12 (%?). Could you please clarify this? For the other increases described... does the same happen?
In general, the article seems relevant to me. It is relatively well built and provides interesting information. However, it lacks important details: with what software did they perform the calculations? the graphics? There are many bibliographic citations missing to support the authors' claims and... variety in the citations. In short, there is still a lot to fix.
Author Response
Dear Reviewer:
Thank you for your feedback on our paper, " Statistical analysis and scenario modeling of non-stationary runoff change in the Loess Plateau: A novel application of GAMLSS model (water-2922860)". Those are excellent and helpful comments. We attentively read over the comments and responded accordingly. We uploaded the revised manuscript file following the directions in your letter. Text revisions are highlighted in red for modification. The following are the responses to the reviewer's comments.
We genuinely appreciate your time and consideration on this revised version of the paper.
**********************************************************************************************************
L55: That "as is well known..." for a statement on national geography...the authors might as well replace such a sentence with: "as demonstrated by X et al, and Y et al"... /major factors in the water cycle process on the Loess Plateau have changed significantly as a result of climate change and human activity, and runoff, as the end product of changes in many elements, is greatly influenced by the compound effects of these changes. Since this is a scientific article and not a lecture, the authors should base their assertions on scientific results which, in all cases, entail a certain "uncertainty". I therefore suggest to the authors that, from now on, they should moderate their claims on the basis of sound scientific publications.
Response: We are grateful to the reviewers for their comments, which identified common issues in our manuscript. During the revision process, we included numerous references to ensure that each important statement had adequate literature to support or explain it.
Due to the introduction have been rewrote, the corresponding content have been revised as follows.
Precipitation, evaporation, water storage, and the underlying surface condition are commonly regarded as the primary causes of watershed runoff. Recently, they have seen varying degrees of change, impacted by increasingly intense climate change and human activities around the world, primarily generating non-stationary changes in runoff in arid and semi-arid regions [1-2]. Furthermore, the elements impact and push each other, amplifying their interactions and driving effects on runoff in these regions [3-5]. The Loess Plateau (LP) is the most common area, and it is receiving increased attention, particularly since the vegetation cover has altered dramatically [6-8]. As widely described in the literature, major elements in the LP's water cycle process have changed dramatically in recent years as a result of climate change and human-induced vegetation regreening [9]. Runoff, as the result of changes in numerous elements, is heavily influenced by the compound impacts of these changes [7, 10-11]. Physically, precipitation, evapotranspiration, and groundwater storage have a direct impact on runoff change, whereas changes in the underlying surface are the primary environmental factors that indirectly cause runoff changes. However, when the streamflow is non-stationary, it is extremely difficult to disentangle the contributions of various elements to the inconsistent change in runoff to figure out the attribution explanations of significant variations in runoff on the LP.
L91: It seems to me that this is the first time that the abbreviation GAMLSS appears here, with the exception of the abstract. Would it be a good idea for the authors, before using the abbreviation, to write down its full meaning? As they have done in previous lines with other abbreviations.
Response: Thank you for your kindly reminder. We have revised it as follows.
The Generalized Additive Model in Location, Scale and Shape (GAMLSS) model is a tool for modeling time series under non-stationary conditions [17].
L99: I still see a certain "sufficiency" on the part of the authors which, for a scientific publication, seems unethical to me.
Response: Thank you for your insightful comments about the lack of rigor in our scientific expression. We have checked and revised the entire text to maximally prevent this imprecise wording.
Although the aforementioned research has made significant progress in interpreting the interactive effects of the hydrological cycle on the LP, as well as analyzing the attribution of non-stationary characteristics in runoff sequences, these two models or methods have some limitations. For example, the time series analysis can only provide average and general contribution estimates, such as from climate change, ecological restoration (ER) strategies, or human activities. In other words, while these methods may be beneficial for analyzing nonstationary time series, their intrinsic static regression aspect does not properly describe many complicated physical interaction processes [15]. Hydrological models, such as lumped (IHACRES), semi-distributed (HEC-HMS), and fully-distributed (SWATgrid), accurately simulate runoff in smaller catchments under most hydroclimatic conditions, but they frequently fail in larger catchments, regardless of hydroclimatic conditions [16], particularly in catchments with non-stationarity in rainfall-runoff relationships. This implies a need for additional research to promote the establishment of the cross-driving relationship between runoff non-stationary changes and their driving elements, as well as the quantitatively evaluation of variations in the driving elements and their impact on runoff alterations, both of which are critical in formulating water resource management policy and addressing water shortages on the LP.
L104-107: It is confusing. Could you rephrase it so that the reader does not have to reread it to understand everything?
Response: Thank you for the kind reminder. We revised this sentence, and the changes are as follows.
This implies a need for additional research to promote the establishment of the cross-driving relationship between runoff non-stationary changes and their driving elements, as well as the quantitatively evaluation of variations in the driving elements and their impact on runoff alterations, both of which are critical in formulating water resource management policy and addressing water shortages on the LP.
L148: Figure 1 has little graphical definition.
Response: Thanks for your kindly reminder. We have revised the Figure 1.
L152-154: This is what the authors claim... Where is it proven, and will they prove it later in this article? Is a bibliographic citation necessary to support such a claim? Check it out!
Response: Thank you for your insightful comments.
We have revised these sentences to express clearly. The modified text is as follows.
The basin's yearly scale water balance formula (P=E+ET+ΔG, when surface water storage changes little) reveals that precipitation, actual evapotranspiration, and groundwater storage variables are the principal drivers of runoff changes. The underlying surface conditions have a significant impact on actual evapotranspiration and groundwater storage variables, making them crucial concerns that cannot be overlooked when investigating the evolution of the runoff process in changing environments.
L154-159: So what now? Now they quote two articles and refer to what they stated in the previous sentences? This way it is not possible to understand what the authors mean. This needs to be rephrased, otherwise readers will not know what they are referring to.
Response: Thanks for your comments. We are very aware of the irrationality expressed in this piece of content. Thus, we revised these sentences.
The basin's yearly scale water balance formula (P=E+ET+ΔG, when surface water storage changes little) reveals that precipitation, actual evapotranspiration, and groundwater storage variables are the principal drivers of runoff changes. The underlying surface conditions have a significant impact on actual evapotranspiration and groundwater storage variables, making them crucial concerns that cannot be overlooked when investigating the evolution of the runoff process in changing environments.
-This whole paragraph at the beginning of the Methodology section is confusing and repetitive of what the authors have said in previous paragraphs. I suggest rephrasing it so that it serves as an "introduction" to "methodology", but expressed in an orderly and simple way.
Response: We appreciate your constructive feedback. We altered the entire paragraph.
The basin's yearly scale water balance formula (P=E+ET+ΔG, when surface water storage changes little) reveals that precipitation, actual evapotranspiration, and groundwater storage variables are the principal drivers of runoff changes. The underlying surface conditions have a significant impact on actual evapotranspiration and groundwater storage variables, making them crucial concerns that cannot be overlooked when investigating the evolution of the runoff process in changing environments. As a result, this study selected precipitation, actual evapotranspiration, groundwater storage variables, and the parameter n in the Budyko equation [26] which represents changes in the underlying surface, as the key driving forces impacting runoff variations. Therein, groundwater storage variables are obtained by the USGS RORA model [27-28]. Also, a GAMLSS inconsistent model is developed by identifying and establishing the interaction of runoff changes and their physical driving elements (e.g., groundwater storage variables, actual evapotranspiration, and underlying surface characteristics) as covariates [29]. This model studies how nonstationary fluctuations in yearly precipitation, groundwater storage factors, and actual evapotranspiration affect annual runoff. Finally, using the planned scenario, runoff variations caused by multiple diverse driving forces were simulated, and alternative co-evolution rules were investigated. Therein, the first year of the entire series (1957) is selected as the base year of the evolution analysis.
L168-184: This whole section is very interesting and relevant. But... Has it been fully developed by the authors, and has no one in the world, or before, contributed anything about it?
Response: Thank you for the affirmation and encouragement.
This is a figure that depicts and explains the cross-relationships we found based on the physical driving features of various variables in the hydrological cycle. As a result, we did not create this interdependence. However, we proposed using statistical approaches (GAMLSS) to quantitatively describe and simulate relationships. There was no analogous research discovered in the author's limited review of the literature.
L188-192: I do not think that the quote provided by the authors is the most appropriate for the technique they are describing. Through a simple web search, any reader can see that, in a much more explicit and extensive way, the skill on GAMLSS can be obtained after reading the more related references. Therefore, I suggest to the authors to be more scientifically ethical and instead of citing only articles of their nationality and interest, cite those who have developed the specific techniques. In addition, they should think of the readers, whose scientific ambition perhaps exceeds the topic the authors themselves are pursuing in this article.
Response: Thanks for your kindly reminder.
We have revised this manuscript and introduce the GAMLSS in two sections. More references are cited in this revision.
Introduction
The Generalized Additive Model in Location, Scale and Shape (GAMLSS) model is a tool for modeling time series under non-stationary conditions [17]. It supports a variety of random variable frequency distribution types, and is extremely useful in constructing linear or nonlinear functional relationships between distribution function position parameters, scale parameters, shape parameters, and explanatory variables [18]. The GAMLSS framework has been widely applied in non-stationary frequency analysis, modeling, and forecasting in hydrology [19-22]. This GAMLSS feature also allows for cross-driving interactions between runoff and driving elements, or between driving elements themselves. In light of this, the goal of this paper is to create an inconsistent hydrological statistical model (GAMLSS) with physical factors as covariates in order to identify the cross-driving relationship between watershed hydrometeorological (precipitation, actual evapotranspiration, groundwater storage variables, and runoff) and surface environmental variables (underlying surface parameters) with inconsistently changing characteristics. It offers a novel approach to inquiry that differs from previously time series analysis and hydrological models in that it statistically analyzes the cross-driving relationship between elements and investigates the causes of runoff decline.
References
- Rigby, R. A., & Stasinopoulos, D. M. (2005). Generalized additive models for location, scale and shape. Journal of the Royal Statistical Society Series C: Applied Statistics, 54(3), 507-554.https://doi.org/10.1111/j.1467-9876.2005.00510.x.
- Serinaldi, F. (2011). Distributional modeling and short-term forecasting of electricity prices by generalized additive models for location, scale and shape. Energy Economics, 33(6), 1216-1226.https://doi.org/10.1016/j.eneco.2011.05.001.
- Scala, P., Cipolla, G., Treppiedi, D., & Noto, L. V. (2022). The Use of GAMLSS Framework for a Non-Stationary Frequency Analysis of Annual Runoff Data over a Mediterranean Area. Water, 14(18), 2848.https://doi.org/10.3390/w14182848.
- He, C., Chen, F., Long, A., Luo, C., & Qiao, C. (2021). Frequency Analysis of Snowmelt Flood Based on GAMLSS Model in Manas River Basin, China. Water, 13(15), 2007.https://doi.org/10.3390/
- Shiau, J. T., & Liu, Y. T. (2021). Nonstationary analyses of the maximum and minimum streamflow in Tamsui River basin, Taiwan. Water, 13(6), 762.https://doi.org/10.3390/w13060762.
- Rashid, M. M., & Beecham, S. (2019). Simulation of streamflow with statistically downscaled daily rainfall using a hybrid of wavelet and GAMLSS models. Hydrological sciences journal, 64(11), 1327-1339.https://doi.org/10.1080/02626667.2019.1630742.
3.2 GAMLSS model
Generalized Additive Models in Location, Scale, and Shape (GAMLSS) is a semi-parametric regression model that analyzes the frequencies of stationary and non-stationary runoff and other features [17-22, 29-30].
References
- Shao, S., Zhang, H., Singh, V. P., Ding, H., Zhang, J., & Wu, Y. (2022). Nonstationary analysis of hydrological drought index in a coupled human-water system: Application of the GAMLSS with meteorological and anthropogenic covariates in the Wuding River basin, China. Journal of Hydrology, 608, 127692. https://doi.org/10.1016/j.jhydrol.2022.127692.
- Scala, P.; Cipolla, G.; Treppiedi, D.; Noto, L.V. (2022). The Use of GAMLSS Framework for a Non-Stationary Frequency Analysis of Annual Runoff Data over a Mediterranean Area. Water, 14, 2848. https://doi.org/10.3390/w14182848.
L227-232: Once again... the values provided by the authors as a reference on the goodness of fit of the model, or the goodness of fit of the model... where do they come from? Is it the authors who determine this validity?
Response: Thanks for your kindly reminder. We have included the appropriate references in the revised draft to explain the rationale of the necessary parameter or coefficient values.
In addition, this study employs the mean, variance, skewness coefficient, kurtosis coefficient, and Filliben correlation coefficient as statistical markers of the residual sequence. The mean is closer to 0, the variance is closer to 1, the skewness coefficient is closer to 0, the kurtosis coefficient is closer to 3, and the Filliben coefficient is closer to 1, indicating that the model fits better [31].
References
- Wang, Y., Duan, L., Tong, X., Liu, T., Li, D., & Li, W. (2023). Non-stationary modeling of wet-season precipitation over the Inner Mongolia section of the Yellow River basin. Theoretical and Applied Climatology, 151(1), 389-405. https://doi.org/10.1007/s00704-022-04279-y.
L338: Figure 3 has such a low definition that it is very difficult to understand it.
Response: Thanks for your kindly reminder.
In the revised manuscript, we included a high-resolution graph and increased the font size on the axis.
L350: Figure 4 has such a low definition that it is very difficult to understand it.
Response: Thanks for your kindly reminder.
In the revised manuscript, we included a high-resolution graph and increased the font size on the axis.
L374: Figure 5 has such a low definition that it is very difficult to understand it.
Response: Thanks for your kindly reminder.
In the revised manuscript, we included a high-resolution graph and increased the font size on the axis.
L397: Figure 6 has such a low definition that it is very difficult to understand it.
Response: Thanks for your kindly reminder.
In the revised manuscript, we included a high-resolution graph and increased the font size on the axis.
L399: Figure 7 has such a low definition that it is very difficult to understand it.
With the rest of the figures it is more or less the same: the definition is so low that you can hardly read the titles of the axes, the levels and the graphs themselves. Anyway
Response: Thanks for your kindly reminder.
In the revised manuscript, we included a high-resolution graph and increased the font size on the axis.
L553-563: Up to the last sentence, the authors summarise in the conclusion section. Then the conclusions begin. Well, I propose to remove all that summary (which is already explained above, and summarised in the abstract itself) and start with the conclusions, after all, this is "their section".
Response: We appreciate your constructive feedback. We altered the start paragraph in the conclusion.
This study investigated the cross-driving relationship between watershed hydrometeorological (precipitation, actual evapotranspiration, groundwater storage variables, and runoff) and surface environ-mental variables (underlying surface parameters) with inconsistently changing characteristics by building a non-consistent GAMLSS for each driving variable with physical factors as covariates. The main findings are as follows:
L565: “The XX-distribution-based GAMLSS model”. We already know very well that it is based on the GAMLSS model, perhaps it is not necessary to repeat it for all the models analyzed.
Response: We appreciate your constructive feedback. We deleted “GAMLSS” in the conclusion.
The GU-distribution-based model with precipitation, actual evapotranspiration, and underlying surface parameters as covariates has a better fitting effect for groundwater storage variables, with a smaller AIC value compared to the consistent model and inconsistent model with time as a covariate. After the year 2000, patterns in time show that actual evapotranspiration and underlying surface parameters in the basin became more important to groundwater storage variables, while provoked groundwater storage variables quickly decreased.
The LO-distribution-based model, which includes precipitation and underlying surface parameters as covariates, has a superior match for actual evapotranspiration, with an AIC value less than the consistent model and inconsistent model, which include time as a covariate. After 2000, the contribution of underlying surface parameters to the actual evapotranspiration significantly increased, leading to an increase in evapotranspiration in the basin.
For yearly underlying surface parameters, the NO-distribution-based model with precipitation as a covariate fits better when both the mean μ and the mean deviation σ change because of physical factors. Additionally, its AIC value is lower than that of a consistent sexual model and a non-uniform model with time as a covariate.
L578-581: It is difficult to understand. I recommend writing it in a simpler way. Furthermore, I suppose that the authors have slipped in a surprising error in the translation: "the AIC value is less than a consistent sexual model"; I suggest you be more careful when translating your scientific interpretation of your results.
Response: We appreciate your constructive feedback. We have revised this sentence.
For yearly underlying surface parameters, the NO-distribution-based model with precipitation as a covariate fits better when both the mean μ and the mean deviation σ change because of physical factors. Additionally, its AIC value is lower than that of a consistent sexual model and a non-uniform model with time as a covariate.
L582-583: Readers already know that. They should reformulate the paragraph so that this, which is repeated in previous paragraphs of the article, is integrated and can be read with the meaning of what the authors wish to conclude next.
Response: We appreciate your constructive feedback. We have removed the first sentence in original version and revised the corresponding text.
The findings in interacting effects revealed that precipitation and groundwater storage variables had positive contributions to runoff variations, whereas actual evapotranspiration had a negative effect. Particularly in 2000, variations in actual evapotranspiration and groundwater storage variables will continue to contribute to runoff changes year after year, resulting in a progressive decrease in runoff.
L589-599: It would seem to the authors to be a good idea to add the percentage increase of the different variables of interest. That is, "if precipitation increases by 10%, the underlying surface parameters increase by 0.12 (%?). Could you please clarify this? For the other increases described... does the same happen?
Response: We appreciate your constructive feedback. We updated the phrasing, and 0.12 simply denotes a change in value.
The scenario simulation results demonstrate that if precipitation increases by 10%, the underlying surface parameters increase the value by 0.12, actual evapotranspiration increases by 38.3 mm, the groundwater storage variable reduces by 0.31 mm, and runoff increases by 1.23 mm. In the situation where the underlying surface characteristics increase by 10%, actual evapotranspiration rises by 18.08 mm, groundwater storage falls by 6.32 mm, and runoff falls by 11.77 mm. In the scenario where precipitation and underlying surface parameters both increase by 10%, the underlying surface parameters increase the value by 0.25, actual evapotranspiration increases by 58.26 mm, groundwater storage variable decreases by 7.28 mm, and runoff decreases by 11.76 mm. It is clear that there are cross-influences across driving variables for runoff, and this influence has time-varying properties, resulting in more complex hydrological impacts and a nonlinear runoff change.
In general, the article seems relevant to me. It is relatively well built and provides interesting information. However, it lacks important details: with what software did they perform the calculations? the graphics? There are many bibliographic citations missing to support the authors' claims and... variety in the citations. In short, there is still a lot to fix.
Response: Thank you for your valuable feedback on our paper.
We carefully studied the comments from the reviewer and made revisions throughout the article, such as correcting typos, supplementing discussions, improving phrasing, and adding references.
Thank you very much for your consideration and comment on our manuscript.

Round 2
Reviewer 1 Report
Comments and Suggestions for Authors
I agree with the publication of the manuscript in its current form.
The authors responded accordingly to my observations.
The only reservation I have regarding the authors' statement, is that the results obtained with the maximum likelihood method have a deviation of only 5.6% compared to those obtained with the method of linear moments (L-moments).
Reviewer 2 Report
Comments and Suggestions for Authors
I am grateful that you have taken my advice into account. I would like to think that, once you have read them, you will have thought that they would improve the paper. For my part, I think the paper is much improved. The low quality of the images of the figures, I suppose it will be due to the quality of the pdf that they send to the reviewers; I would like to think that when it is published, the figures will have a better "definition". I liked the paper very much.